# Scaf1 promotes respiratory supercomplexes and metabolic efficiency in zebrafish

Carolina García-Poyatos[1,2] (ID), Sara Cogliati[1,3], Enrique Calvo[1], Pablo Hernansanz-Agustín[1], Sylviane Lagarrigue[4], Ricardo Magni[1], Marius Botos[2], Xavier Langa[2], Francesca Amati[4], Jesús Vázquez[1,5], Nadia Mercader[1,2]* (ID) & José Antonio Enríquez[1,6,**] (ID)

## Abstract

The oxidative phosphorylation (OXPHOS) system is a dynamic system in which the respiratory complexes coexist with super-assembled quaternary structures called supercomplexes (SCs). The physiological role of SCs is still disputed. Here, we used zebrafish to study the relevance of respiratory SCs. We combined immunodetection analysis and deep data-independent proteomics to characterize these structures and found similar SCs to those described in mice, as well as novel SCs including $III_2 + IV_2$, $I + IV$, and $I + III_2 + IV_2$. To study the physiological role of SCs, we generated two null allele zebrafish lines for supercomplex assembly factor 1 (scaf1). $scaf1^{-/-}$ fish displayed altered OXPHOS activity due to the disrupted interaction of complexes III and IV. $scaf1^{-/-}$ fish were smaller in size and showed abnormal fat deposition and decreased female fertility. These physiological phenotypes were rescued by doubling the food supply, which correlated with improved bioenergetics and alterations in the metabolic gene expression program. These results reveal that SC assembly by Scaf1 modulates OXPHOS efficiency and allows the optimization of metabolic resources.

**Keywords** metabolism; mitochondria; OXPHOS super-assembly; SCAF1/COX7A2L; zebrafish

**Subject Categoryies** Membrane & Trafficking; Metabolism

**EMBO reports (2020) 21: e50287**

## Introduction

In the last 2 years, the focus of investigation on the structure of the mitochondrial electron transport chain (ETC) has shifted from the dispute over the existence of supercomplexes (SCs) to their putative functional role. In mammals, the best understood mechanism of respiratory complex super-assembly is the interaction between complexes III (CIII) and IV (CIV) mediated by supercomplex assembly factor 1 (SCAF1/COX7A2L) [1]. The carboxy-terminus of SCAF1 is very similar to that of the CIV subunit COX7A2 and replaces it in the subset of CIV molecules that super-assemble with CIII [2]. After some initial doubts [3], which were later dispelled [4,5], the role of SCAF1 in the super-assembly of CIII and CIV is now generally accepted [2]. The process of super-assembly between CI and CIII and CI and CIV to form the respirasome is unknown, but the proposed existence of I + IV SCs [6] suggests that CI-CIII and CI-CIV super-assembly might occur independent from CIII and CIV assembly [7,8]. So far, the interaction between CI and CIV has been mostly studied in SCs containing CI, CIII, and CIV (also named respirasomes). Several forms of respirasomes (SC I + $III_2$ + IV) migrate closely together in blue native gel electrophoresis (BNGE), although the reason for their different apparent molecular weights remains unknown. Even though SCAF1 loss of function abolishes the interaction between CIII and CIV, the current consensus is that the absence of functional SCAF1 does not completely disrupt SC I + $III_2$ + IV formation. However, SCAF1 loss of function strongly reduced the stability and variety of respirasomes [1,2,4].

The super-assembly between CI and CIII was proposed to allow partitioning of coenzyme Q (CoQ) into two communicated functional pools: one trapped in SCs and the other free within the inner mitochondrial membrane [9]. The super-assembly between CIII and CIV allows the control of available CIV through compartmentalization. Both functions optimize the metabolic flux, preventing an electron traffic jam [1] and minimizing reactive oxygen species (ROS) production [10] while maintaining an efficient energy production [9]. However, studies performed on fragmented sub-mitochondrial particles generated by disruption of mitochondrial membranes with detergents challenged this model [11]. These studies concluded that CoQ pools are continuously intermixed at a rate that rules out the possibility of preferential use of CoQ within SCs. Accordingly, these

1  Centro Nacional de Investigaciones Cardiovasculares Carlos III, Madrid, Spain
2  Institute of Anatomy, University of Bern, Bern, Switzerland
3  Instituto de Nutrición y Tecnología de los Alimentos (INYTA), Universidad de Granada, Granada, Spain
4  Aging and Muscle Metabolism Laboratory, Department of Biomedical Sciences, University of Lausanne, Lausanne, Switzerland
5  CIBERCV, Madrid, Spain
6  CIBERFES, Madrid, Spain
  *Corresponding author. Tel: +41316318477; E-mail: nadia.mercader@ana.unibe.ch
  **Corresponding author. Tel: +34 914531265; E-mail: jaenriquez@cnic.es

studies defended the notion that the super-assembly between CI and CIII in the form of SC I + III$_2$ or SC I + III$_2$ + IV would lack any bioenergetic role [5]. A very recent publication analyzing isolated SC I + III$_2$ also supports the model were partitioning of CoQ into SC I + III$_2$ has functional implications in the oxidation of NADH [12].

*In vivo*, the functional role of SCs has been supported by the positive correlation of the proportion of respirasomes (SC I + III$_2$ + IV) with exercise [13] and with lower mitochondrial ROS production [14–16]. Interestingly, some common mouse strains, such as C57BL/6, have a non-functional SCAF1 protein (SCAF1[111]) lacking two amino acids as compared with functional SCAF1 (SCAF1[113]). SCAF1[111] mice do not assemble SC III$_2$ + IV, and the formation of respirasomes is affected [1,2]. Since SCAF1[111] mice do not present apparently any specific phenotype compared to SCAF1[113] mice, this has contributed to the arguments against a physiological role for the super-assembly of CIII and CIV. Furthermore, a recent study performed in cultured cells suggested that the ablation of SCAF1 has no bioenergetic relevance [17], a conclusion that was later refuted by the demonstration that SCAF1 was critical for the cellular metabolic adaptation to the increase in energy demands associated with endoplasmic reticulum and nutrient stresses [18]. In addition, tumorigenic capacity, mediated by enhanced hypoxia resistance in human cancer models, has been proposed to be dependent on CIII and CIV super-assembly [19]. Controversy over the bioenergetic role of CIII and CIV super-assembly thus remains, and a physiological role is still unclear. Here, we studied respiratory SCs in zebrafish (*Danio rerio*) and evaluated the super organization and plasticity of the oxidative phosphorylation (OXPHOS) system. We identified SCs as those previously described in mice, such as SC I + III, SC III$_2$ + IV, and SC I + III$_2$ + IV, but also novel SCs, including SC III$_2$ + IV$_2$, SC I + IV, and SC I + III$_2$ + IV$_2$. With the aim of studying the bioenergetic and physiological consequences of impaired super-assembly, we generated *scaf1* null mutant zebrafish lines. *scaf1*$^{-/-}$ fish showed a disrupted interaction of CIII and CIV, and presented a prominent phenotype, including a smaller size and decreased fertility. Bioenergetics analysis revealed that *scaf1* ablation promotes an inefficient OXPHOS capacity due to the disruption of the compartmentalization of CIV. Strikingly, phenotypic alterations in *scaf1*$^{-/-}$ animals are fully corrected by doubling the food supply but not by changing the regime to a high-fat diet. The phenotypic rescue occurred in the absence of a recovery of OXPHOS super-assembly and correlated with alterations in the metabolic gene expression program.

Overall, these results confirm a role for SCs in the efficiency of the respiratory chain in a vertebrate animal model and reveal that SCs provide an advantage in the optimization of metabolic resources.

## Results

### OXPHOS super-assembly in zebrafish

To re-evaluate whether the disruption of SCs formation has any impact at the organismal level, here we used the zebrafish animal model. OXPHOS genes have been reported to be highly conserved along evolution [20], and respiratory complexes have been shown to super-assemble during zebrafish development [21]. Thus, we first characterized the pattern of respiratory complex super-assembly in adult zebrafish using 1D and two-dimensional (2D) BNGE as well as Blue-DiS-based proteomics. Purified zebrafish skeletal muscle mitochondria were run in parallel with mitochondria from mouse skeletal muscle of either functional SCAF1[113] (CD1) or non-functional SCAF1[111] (C57BL/6J) mice (Fig 1A–D). As expected, the SC III$_2$ + IV SC was absent in SCAF1[111] mitochondria and the abundance of the respirasome (SC I + III$_2$ + IV) was low. The BNGE pattern of zebrafish CI and CIII was similar to that of SCAF1[113] mice, where free I, III$_2$, and SC I + III$_2$ could be easily identified (Figs 1A and EV1A and B to see the split channels of Fig 1A). The migration pattern of zebrafish CIV paralleled that of mice and revealed the presence of free CIV, IV$_2$, and SC III$_2$ + IV in zebrafish. It also confirmed the presence of the respirasome, which migrated faster than that of mice (Figs 1B and EV1C to see CIV split channel of Fig 1B). We also noted some conspicuous differences between the pattern in zebrafish and the classical mouse pattern. Specifically, we observed a co-migration of CIII and CIV just above free CI (Figs 1B and EV1C to see CIV split channel of Fig 1B), as well as co-migration of CI and CIV just below SC I + III$_2$, which is compatible with the interaction of CI and CIV. The CI and CIV interaction was barely visible by immunodetection (Fig EV1A and C), but could be clearly revealed by CI (Fig 1C) and CIV (Fig 1D) in-gel activity, which also confirmed the observations based on immunodetection.

The novel band containing CIII and CIV is compatible with the presence of a dimer of CIII with two molecules of CIV. Two alternative structural arrangements could generate this band—the interaction of two dimers, III$_2$ and IV$_2$ (SC III$_2$ + IV$_2$), or the interaction of one dimer of III with two monomers of IV (SC IV + III$_2$ + IV or SC 2IV + III$_2$). To distinguish between these possibilities, we performed 2D BNGE using digitonin as a detergent in the first dimension to preserve the integrity of SCs, and *n*-dodecyl-β-D-maltoside (DDM) in the second dimension to disaggregate SCs while substantially preserving the integrity of the complexes [22] (Fig EV1D and E). This analysis allowed us to differentiate between co-migration and true DDM-sensitive interactions and to discern whether CIV was in a monomeric or dimeric form. Although DDM partially disassembled IV$_2$ into its monomers, we were still able to

**Figure 1. OXPHOS super-assembly in zebrafish.**

A–D Blue native gel electrophoresis (BNGE) of mouse (M) C57BL/6J (111), CD1 (113), and zebrafish (ZF) skeletal muscle digitonin-solubilized mitochondria. (A, B) Immunodetection of the indicated proteins after BNGE, (C) in-gel activity for CI and (D) CIV (shown is a representative gel from two technical and two biological replicates).

E–H BNGE of whole-body zebrafish digitonin-solubilized mitochondria of *scaf1*$^{\Delta 1/\Delta 1}$ (−/−) and its respective *scaf1*$^{+/+}$ counterpart. (E, F) Immunodetection of the indicated proteins, (G) in-gel activity of CI and (H) CIV (representative gel from two technical and three biological replicates).

I 2D BNGE/SDS electrophoresis: 1$^{st}$ dimension with digitonin (Dig) and 2$^{nd}$ dimension with SDS, followed by immunoblotting with the indicated antibodies to identify the proteins detected by the commercial anti-SCAF1 antibody. Asterisks indicate missing bands in *scaf1*$^{\Delta 1/\Delta 1}$.

   

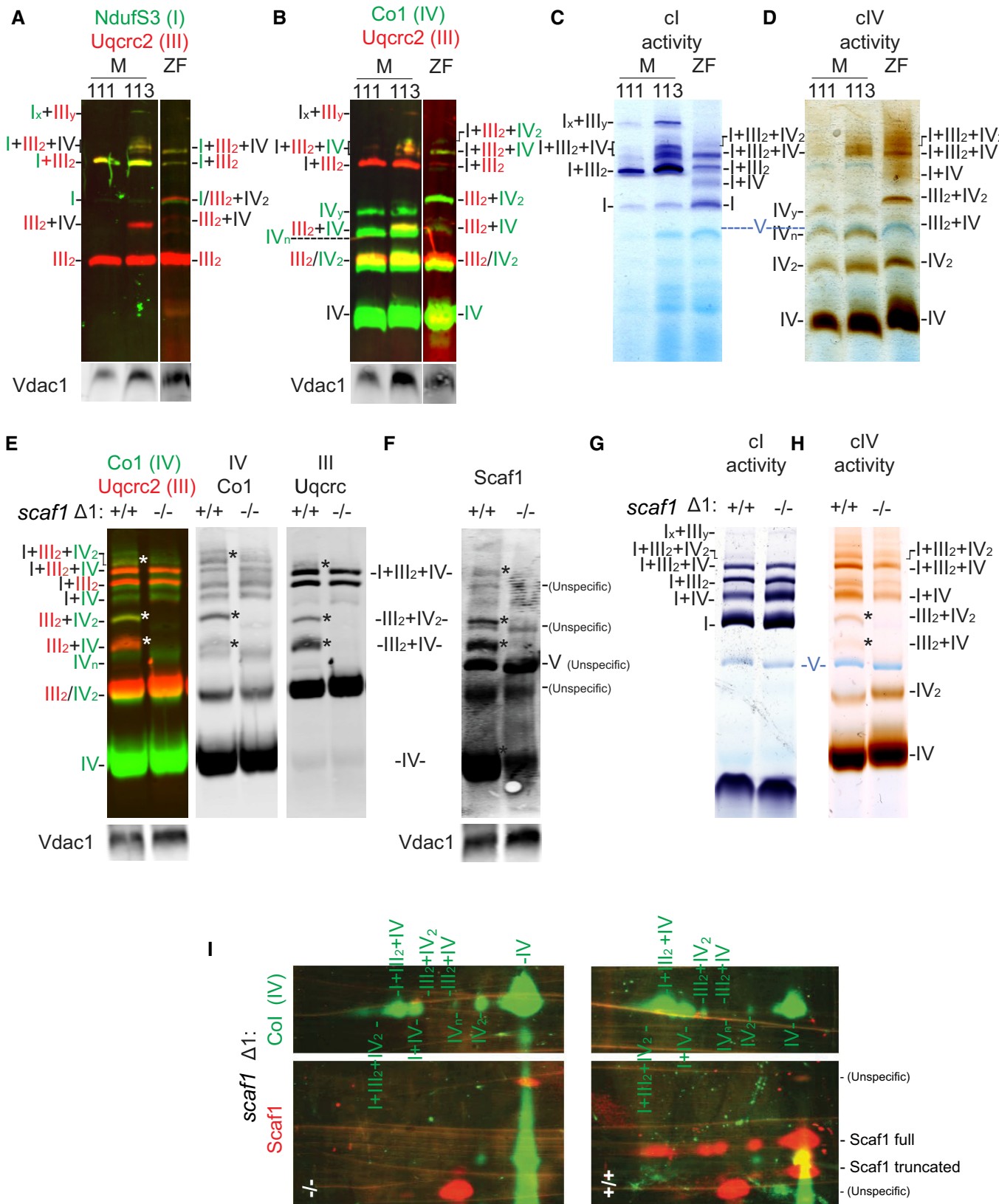

**Figure 1.**

detect a considerable proportion of $IV_2$. Indeed, we found a dimer of CIV ($IV_2$) present in the novel high molecular weight band containing CIII and CIV, indicating that this co-migration is due to the physical interaction between $III_2$ and $IV_2$ (Fig EV1D and E). In addition, 2D-BNGE analysis demonstrated that CI and CIV co-migration was due to a physical interaction between them in the form of SC I + IV, which could be disrupted by DDM (Fig EV1D and E). Interestingly, this assay also revealed the presence of CIV dimers in high molecular weight structures associated with CI and CIII (putative SC I + $III_2$ + $IV_2$) or with CI alone (putative SC I + $IV_2$; Fig EV1D and E).

These observations were confirmed in mitochondria purified from whole zebrafish (Fig EV2A–E). Of relevance, the pattern of zebrafish BNGE bands was stable at 4 and 10 g/g proportion of digitonin, whereas 1 g/g proportion was unable to sufficiently solubilize the mitochondrial membranes (Fig EV2F).

To evaluate whether the proposed role of respiratory complex super-assembly plasticity during the adaptation to variations in metabolic resources applies to zebrafish, we fed adult zebrafish for 6 weeks with a low-protein/low-fat (LP/LF) diet, which leads to malnutrition (Fig EV2G–I). LP/LF-fed fish showed a decrease in weight, which was accompanied by an increase in SC I + $III_2$ and SC I + $III_2$ + $IV_{1-2}$ and a decrease in SC $III_2$ + $IV_1$, while SC $III_2$ + $IV_2$ remains stable (Fig EV2I). These results confirm that the organization of the ETC is modulated in response to metabolic changes also in zebrafish.

Given the similarities with mice and the possibility to study novel SCs such as I + IV and those containing $IV_2$, we conclude that the zebrafish is a valuable model to study SC assembly and function.

## Role of Scaf1 in zebrafish respiratory complexes super-assembly

To assess structural and physiological effects of alterations in complex super-assembly, we generated a zebrafish *scaf1* null mutant model. Zebrafish *scaf1*, also known as *cox7a2l* and *cox7a3*, is highly conserved compared with mammals (Appendix Fig S1A). We generated two independent *scaf1* zebrafish null allele lines using CRISPR/Cas9 technology (Δ1 and Δ2; Appendix Fig S1B and C). We introduced premature STOP codons after amino acids 43 and 51, respectively, which, according to sequence information, lead to a short non-functional protein (Appendix Fig S1D and E). As anticipated, we could not detect Scaf1 protein by Western blotting of *scaf1*$^{\Delta 1}$ and *scaf1*$^{\Delta 2}$ homozygous fish (Appendix Fig S1F). We next extracted mitochondria from whole *scaf1*$^{+/+}$ and *scaf1*$^{-/-}$ fish and compared SC formation by BNGE (Fig 1E–H). The lack of Scaf1 completely eliminated the two bands of SCs containing CIII and CIV (SC $III_2$ + IV, SC $III_2$ + $IV_2$). It also strongly diminished the SC

I + $III_2$ + $IV_2$ band, whereas the SC I + $III_2$ + IV was only slightly decreased in intensity. Immunodetection of CIII, CIV, and Scaf1 (Fig 1E and F) as well as CI and CIV in-gel activity assays (Fig 1G and H) confirmed these observations.

Scaf1 was immunodetected in BNGE in a greater number of bands than those reported in mammals, from which three completely disappeared in *scaf1*$^{-/-}$ samples (Fig 1F). We found that this is due to the use of a new Scaf1 antibody generated against the entire protein, as the previously well-established antibody generated against a Scaf1-specific peptide was discontinued. The new antibody caused the appearance of strong non-specific immunoreactive bands. To distinguish between Scaf1-specific and non-specific immunodetected signals, we performed 2D-BNGE/SDS–PAGE electrophoresis of *scaf1*$^{+/+}$ and *scaf1*$^{-/-}$ samples (Fig 1I). This analysis revealed that the antibody recognized four proteins of different molecular weight. Two of them remained present in the *scaf1*$^{-/-}$ sample, indicating that they correspond to spurious signals. From the two specific signals that disappeared in *scaf1*$^{-/-}$ samples, one matched with the expected size of Scaf1 and migrated in 5 different spots corresponding to complexes IV, $IV_2$, SC $III_2$ + IV, SC $III_2$ + $IV_2$, SC I + $III_2$ + IV, and SC I + $III_2$ + $IV_2$. Unexpectedly, a second specific protein with lower molecular weight was present in two spots (IV, $IV_2$), suggesting the presence of a shorter form of Scaf1 (Fig 1I).

To further study the effect of Scaf1 loss of function on SC assembly, we performed high-throughput Blue-DiS-based proteomics of the entire BNGE run of *scaf1*$^{+/+}$ and *scaf1*$^{-/-}$ samples. This allowed the detection of around 9,000 UNIPROT entries (Dataset EV1). In the currently annotated zebrafish proteome, most of the UNIPROT entries are unreviewed, a large proportion is referred to as fragments, and a great number of entries correspond to unidentified proteins (Dataset EV1). Therefore, some entries may correspond to the same gene and would not accurately report the number of different proteins within the gel. Proteins were detected at a very variable concentration covering six orders of magnitude, from which mitochondrial proteins, and in particular the inner membrane proteins, were the most abundant (Dataset EV1). Proteomic analysis confirmed the pattern of OXPHOS complexes obtained by immune analysis and in-gel activity (Figs 2A and EV3C and D). It also revealed the high reserve of the CI N-module retaining its NADH dehydrogenase activity (Figs 1G and EV3C and D). This accumulation was paralleled by a significant amount of CI subcomplex lacking NADH activity that co-migrated with SC $III_2$ + IV (Figs 1G and EV3C and D). Of note, proteomic analysis also confirmed the loss of Scaf1 protein in *scaf1*$^{-/-}$ (Fig EV3C and D).

The proteomic analysis also allowed a more quantitative estimation of the changes induced by Scaf1 ablation (Fig EV3E–G). It confirmed that the absence of functional Scaf1 impaired the super-assembly of CIII with CIV (Fig EV3E) and showed that both CIII and

**Figure 2. Blue-DiS proteomics of *scaf1*$^{+/+}$ and *scaf1*$^{-/-}$ isolated mitochondria.**

A   Quantitative data-independent scanning (DiS) mass spectrometry protein profiles for CI, CIII, and CIV. Vertical numbers indicate the BNGE gel slices. Left and right profiles correspond to *scaf1*$^{+/+}$ and *scaf1*$^{\Delta 1/\Delta 1}$ animals, respectively. Red heatmap corresponds to the *E*-score from two proteotypic Scaf1-derived tryptic peptides spanning sequences ascribed to the CIII-interacting site (in green) and to the CIV-interacting site (in yellow) in *scaf1*$^{+/+}$ fish. Thick blue line, marked with an asterisk, indicates the putative proteolytic site in Scaf1.

B   Sequence alignment of Scaf1 protein in mouse and zebrafish. Structural and functional regions previously described in mouse are indicated in shaded gray boxes. Thick blue line indicates the proteolytic processing site in the mouse sequence.

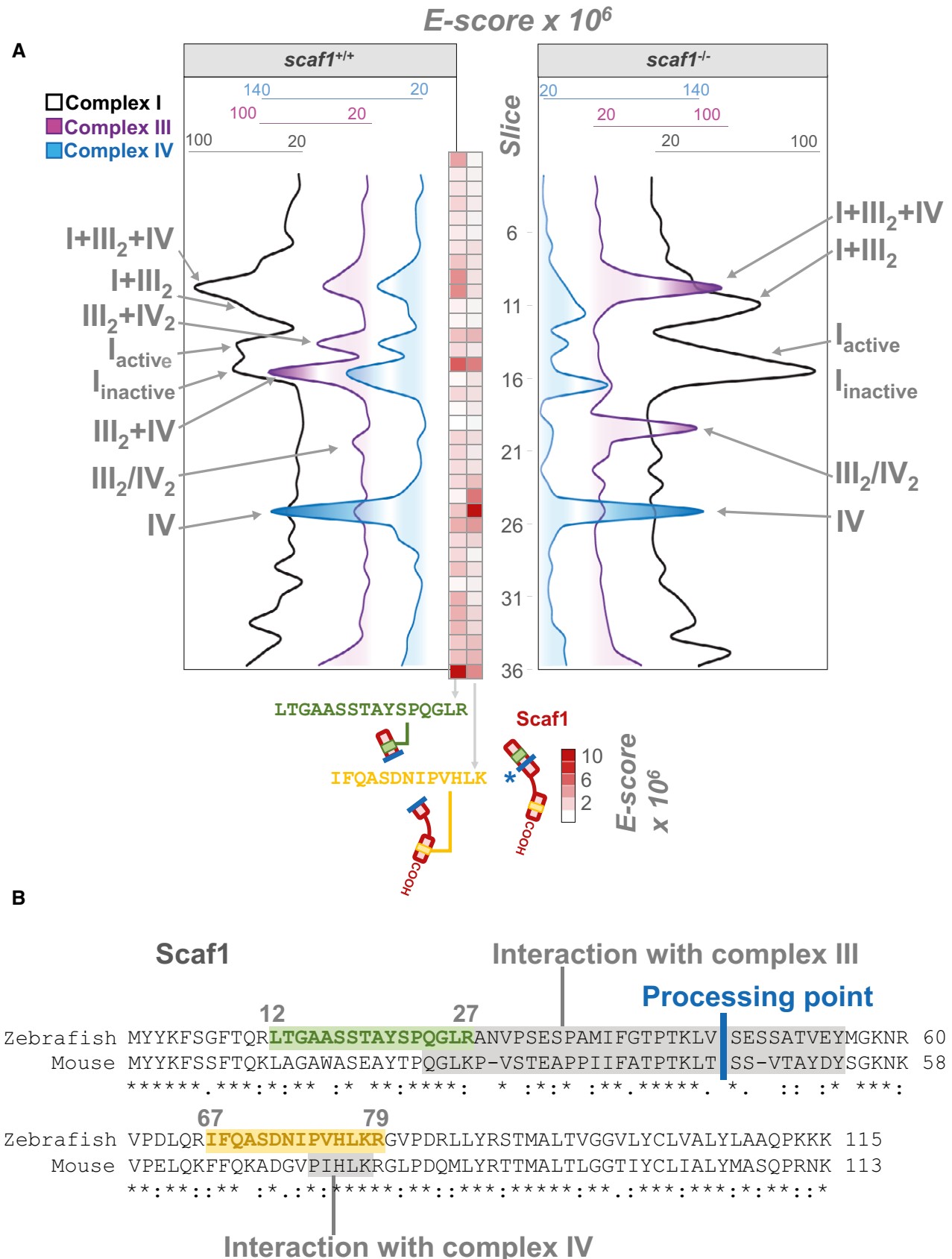

**Figure 2.**

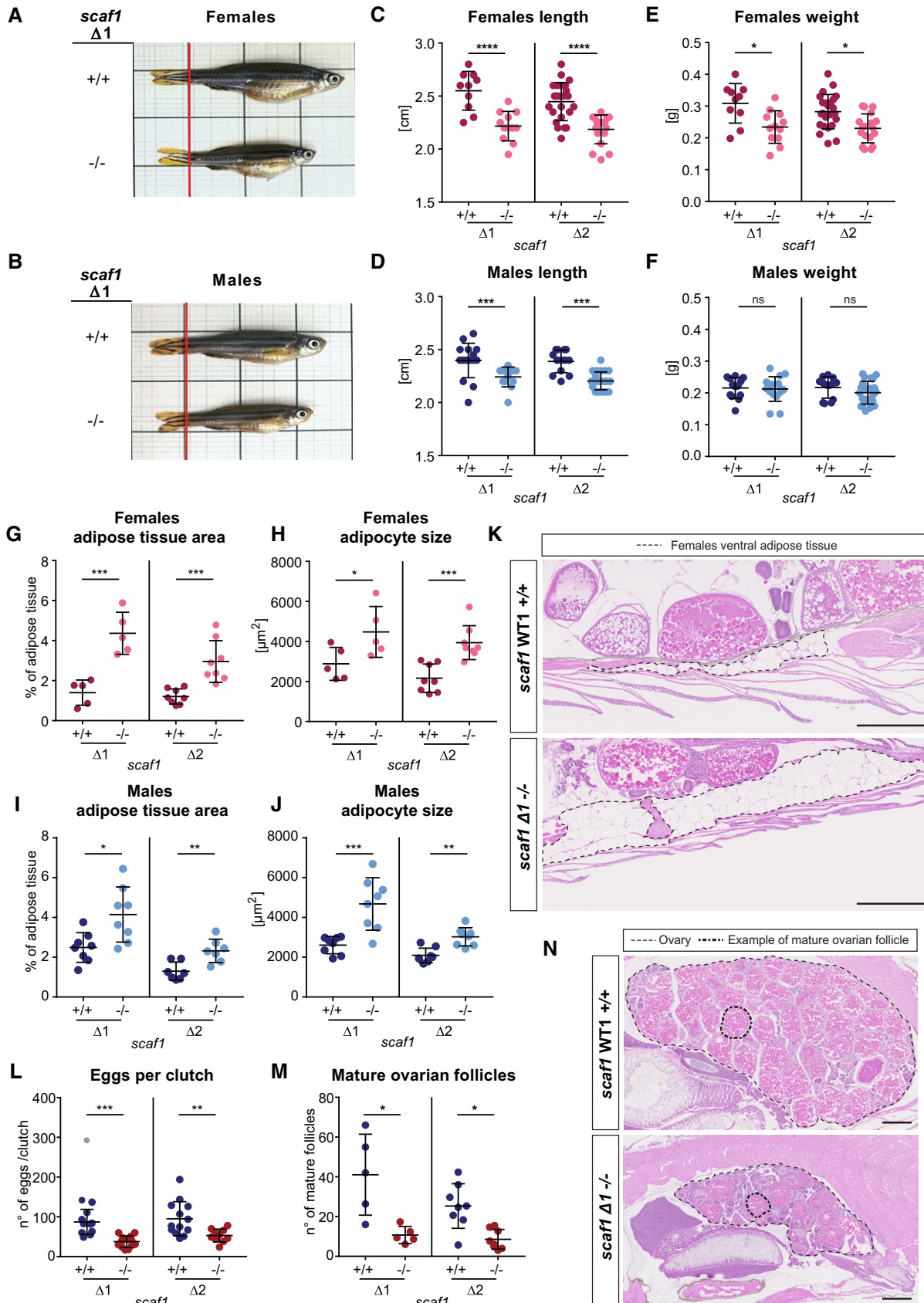

Figure 3.

**Figure 3. Phenotype consequences of Scaf1 loss of function.**

A, B Representative images from $scaf1^{+/+}$ and $scaf1^{-/-}$ (A) female and (B) male adult zebrafish.

C–F Size of $scaf1^{\Delta1/\Delta1}$ and $scaf1^{\Delta2/\Delta2}$ ($scaf1^{-/-}$) fish in comparison with their respective $scaf1^{+/+}$ wild type (WT) lines, (C) length and (E) weight of females ($\Delta1$ +/+ $n = 10$, $\Delta1$ −/− $n = 12$, $\Delta2$ +/+ $n = 24$, $\Delta2$ −/− $n = 18$); (D) length and (F) weight of males ($\Delta1$ +/+ $n = 16$, $\Delta1$ −/− $n = 13$, $\Delta2$ +/+ $n = 13$, $\Delta2$ −/− $n = 23$).

G–K Adipose tissue measurements on hematoxylin–eosin (H&E)-stained adult zebrafish sagittal sections. (G, I) Adipose tissue area per total section area (average of 3 sections/biological replicate) and (H, J) adipocyte size (average of 20–30 adipocytes of ventral adipose tissue per biological replicate) of females (G, H) ($\Delta1$ $n = 5$, $\Delta2$ $n = 8$, same number of animals for homozygous mutants and controls) and males (I, J) ($\Delta1$ $n = 8$, $\Delta2$ $n = 7$, same number of animals for homozygous mutants and controls). (K) Representative images of ventral fat deposits in females (dotted lines).

L–N Effect of Scaf1 loss of function on female fertility. (L) Number of eggs per clutch ($\Delta1$ +/+ $n = 12$, $\Delta1$ −/− $n = 13$, $\Delta2$ +/+ $n = 13$, $\Delta2$ −/− $n = 10$). (M) Quantification of mature ovary follicles per ovary section (average of three sections/biological replicate; $\Delta1$ $n = 5$, $\Delta2$ $n = 8$; same number of animals for homozygous $scaf1^{+/+}$ and $scaf1^{-/-}$). (N) Representative images of H&E-stained ovaries. Dotted lines delineate adipose tissue.

Data information: One-way ANOVA. Outliers are shown in gray and were not considered for the statistical analysis. Data are represented as mean $\pm$ SD. *$P < 0.05$, **$P < 0.01$, ***$P < 0.001$, ****$P < 0.0001$. Scale bars = 500 μm.

CIV completely disappeared from SC III$_2$ + IV$_2$ and SC III$_2$ + IV bands, with the concomitant increase in III$_2$ and IV. Conversely, the electrophoretic mobility of the full and partially assembled forms of CI was unaffected, demonstrating that they co-migrate but do not interact (Fig EV3E–G). Nevertheless, we noticed a significant quantitative shift from the full CI to the partially assembled form of CI and a parallel increase in the amount of the free CI N-module. Proteomic analysis also confirmed that the different forms of respirasomes were still present in the $scaf1^{-/-}$ samples. However, it revealed quantitative differences when comparing $scaf1^{-/-}$ and controls, with a reduction in SC I + III$_2$ + IV and in the putative SC I$_2$ + III$_2$ + IV$_2$, as well as an increase in SC I + III$_2$, the putative SC I$_2$ + III$_2$, and the amount of SC I + IV (Fig EV3E–G).

Proteomic analysis confirmed that the smaller protein co-migrating with IV and IV$_2$ was indeed a version of Scaf1 missing its amino terminus (Fig 2A). By monitoring the presence of two proteotypic peptides for zebrafish Scaf1 located either at the amino- (CIII-interacting domain) or carboxy- (CIV-interacting domain) portions of the protein (Fig 2B), we were able to determine whether Scaf1 was present in full or truncated forms. We found that the proportion of the two peptides was similar (slice 15, red-scaled heatmap in Fig 2A), demonstrating the presence of full-length Scaf1 only in the bands where CIII and CIV interact. By contrast, when Scaf1 was found in free CIV (slice 25, red-scaled heatmap in Fig 2A), the abundance of the amino-peptide was severely decreased and only the carboxy-peptide was detected in significant amounts. We speculate that the short Scaf1 form might derive from the proteolytic cleavage of full-length Scaf1 in a position located in the sequence of interaction with CIII (processing point Fig 2B), disrupting the interaction between CIII and CIV and giving rise to the free complexes. This interpretation is consistent with the fact that the Scaf1 amino terminus appeared at the bottom of the gel (Fig 2A) and with the observation of a similar phenomenon in mouse mitochondria [23].

### $scaf1$ loss of function leads to non-pathological physiological changes in zebrafish

Interestingly, the ablation of Scaf1 caused a prominent phenotype in zebrafish (Fig 3). Both males and females were significantly shorter (Fig 3A–D), and females weighed significantly less (Fig 3E and F).

We further observed that $scaf1^{-/-}$ animals accumulated more adipose tissue and showed increased adipocyte cell size (Fig 3G–K), indicating an altered metabolism. In addition, $scaf1^{-/-}$ females showed a decrease in fertility, with a lower number of eggs per

clutch (Fig 3L), possibly due to delayed oocyte maturation (Fig 3M and N). Despite the observed defect in fertility, embryonic development was normal in $scaf1^{-/-}$ (Appendix Fig S2A–D). Change in length, as seen in adults, became apparent at 3 days post-fertilization (dpf) and reached a stable value at 3 months post-fertilization (mpf) (Appendix Fig S25E–G). Changes in body weight between $scaf1^{-/-}$ and wild-type sibling reached significance after 3 mpf in females and 5 mpf in males (Appendix Fig S2G). Length and body weight alterations, as well as a loss of super-assembly between CIII and CIV, were observed in both $scaf1^{-/-}$ zebrafish lines, $\Delta1$ and $\Delta2$ but not in heterozygous $scaf1^{+/-}$ siblings (Appendix Fig S2H–M). Moreover, $scaf1^{+/-}$ offspring from homozygous $scaf1^{-/-}$ females did not present any phenotype, showing that there is no maternally contributed role for Scaf1 (Appendix Fig S2H–L).

To fully confirm that the observed phenotype is directly caused by the absence of Scaf1, we generated a transgenic Scaf1 gain of function model in the $scaf1^{-/-}$ background (Fig EV4). This transgenic zebrafish line $Tg(ubi:scaf1)$ expresses Scaf1 under the control of the $ubiquitin$ promoter ($ubi$), which drives an ubiquitous expression in the whole zebrafish throughout all developmental stages. $scaf1^{-/-,\ Tg/-}$ recovered the SC III$_2$ + CIV$_{1-2}$ super-assembly (Fig EV4B) and presented increased body size when compared to their non-transgenic $scaf1^{-/-}$ siblings under regular feeding. No differences in size of $scaf1^{-/-,\ Tg/-}$ fish were observed compared to $scaf1^{+/+}$ fish (Fig EV4C–F), and $scaf1^{-/-,\ Tg/-}$ fish were significantly larger than $scaf1^{-/-}$ siblings. This result confirms that the lack of CIII and CIV super-assembly as well as the effect on organismal growth observed in $scaf1^{-/-}$ is specific to loss of Scaf1 function.

In sum, the lack of Scaf1 leads to phenotypes resembling malnourishment in zebrafish (reduced growth and reproduction efficiency) [24,25] when they are fed in equal conditions as $scaf1^{+/+}$ fish. Therefore, Scaf1 loss of function impairs the proper energy conversion from nutrients, storing them in abnormally high amounts at the adipose tissue instead of using them for energy production. Importantly, these results reveal a physiological role for Scaf1 as an OXPHOS supercomplex assembly factor at the organismal level.

### Mitochondrial consequences of $scaf1$ loss of function

To determine whether the observed phenotypes could be directly related to OXPHOS function, we first used transmission electron microscopy (TEM) to establish whether the absence of Scaf1 induced any ultrastructural alterations in zebrafish mitochondria (Fig 4A). We observed more fragmented mitochondria (Fig 4B) with

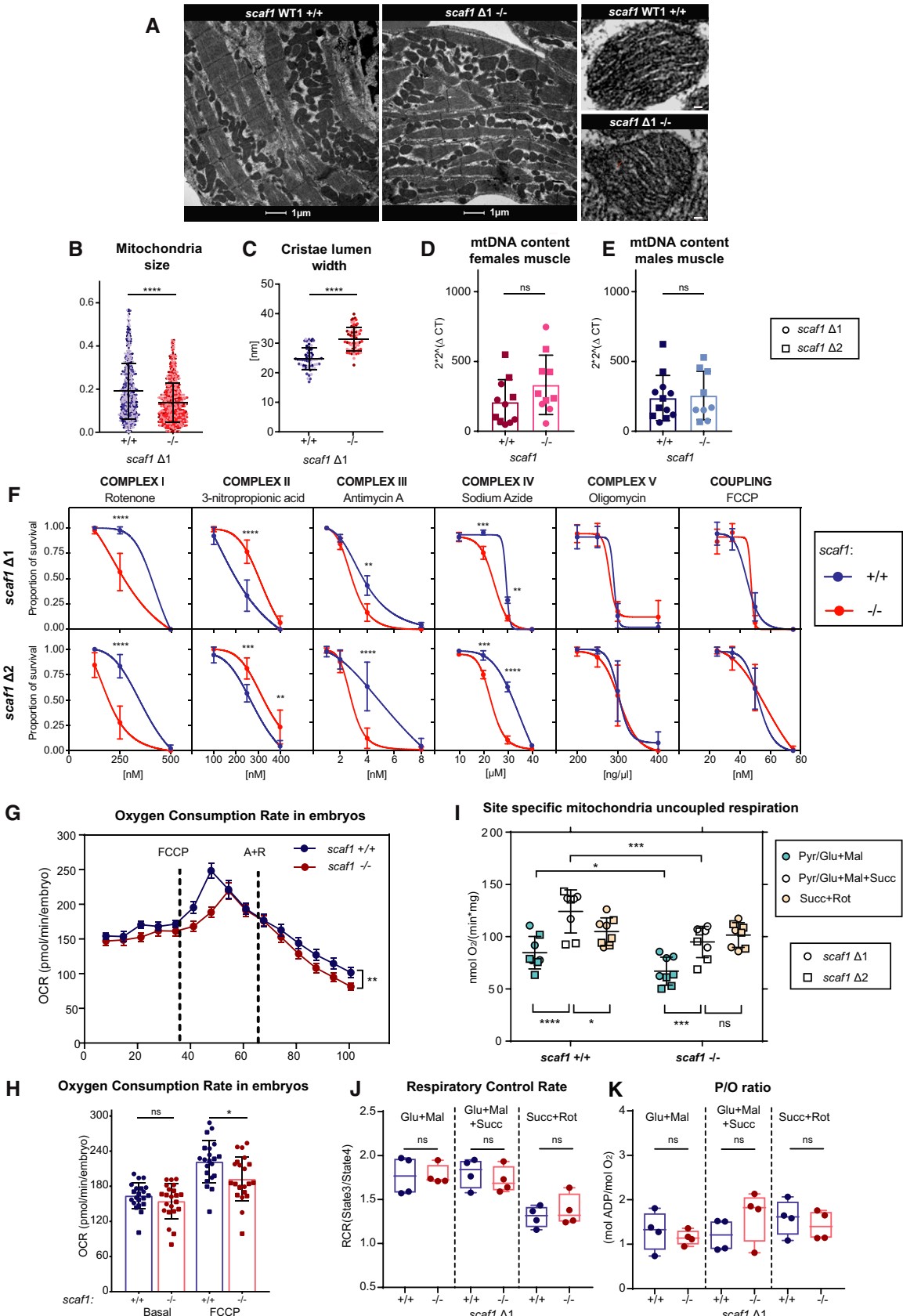

**Figure 4.**

**Figure 4.  Scaf1 loss of function leads to alterations in mitochondrial structure and performance.**

A–C  Transmission electron microscopy image of cardiac muscle from $scaf1^{\Delta1/\Delta1}$ ($n$ = 3) and $scaf1^{+/+}$ fish ($n$ = 3). (A) Representative images showing mitochondria. (B) Mitochondria size (100–150 mitochondria per biological sample). (C) Cristae lumen width (average of three cristae per mitochondria, 20 mitochondria per biological sample). Different biological replicates are represented with different color tones.

D, E  Mitochondrial DNA copy number per nuclear copy number in muscle in females (D) and males (E) ($\Delta1$ $n$ = 6, $\Delta2$ $n$ = 6 and same number for their respective controls).

F  Survival curve of 4 days post-fertilization embryos treated with different concentrations of the indicated OXPHOS inhibitors (three experimental replicates per biological replicate and three biological replicates).

G, H  Oxygen consumption of 48 h post-fertilization embryos using the XFe24 Seahorse analyzer, (G) representative oxygen consumption rate (OCR) profile along time, and (H) maximum OCR ($\Delta1$ $n$ = 11, $\Delta2$ $n$ = 11, $n$ = 10, $n$ = 11, respectively, for their controls).

I  Maximum uncoupled (FCCP) OCR in isolated mitochondria from adult fish (male and females $\Delta1$ $n$ = 4 and $\Delta2$ $n$ = 4, and same number for their respective controls) with the indicated site I [pyruvate (Pyr), glutamate (Glu), malate (Mal)] or site II [succinate (Succ)] substrates.

J, K  Respiratory control ratio (RCR; State 3/State 4) (J) and P/O ratio (K) in isolated mitochondria from adult fish (male and females $\Delta1$ $n$ = 4, and same number for their respective controls) with the indicated substrates.

Data information: (B–E, G, H, J, K) Unpaired $t$-test, (F) two-way ANOVA, and Sidak's multiple comparison test. (I) Two-way ANOVA, *post hoc* Fisher's LSD test. ns $P > 0.05$, *$P < 0.05$, **$P < 0.01$, ***$P < 0.001$, ****$P < 0.0001$. Data are represented as mean ± SD, or (G) as ± SEM. Scale bars = large image 1 μm, small image 50 nm.

wider cristae (Fig 4C). Increased cristae lumen width has previously been associated with the reduction of super-assembly of respiratory complexes in models of cristae junction disruption [26,27]. No differences in mitochondria content were noted, as measured by the ratio of mitochondrial DNA to nuclear DNA (Fig 4D and E).

To further assess whether OXPHOS function was affected, we determined the susceptibility of 4 dpf $scaf1^{+/+}$ and $scaf1^{-/-}$ larvae to pharmacological inhibitors of the respiratory complexes I, II, III, and IV, as well as inhibitors of $H^+$-ATPase (CV) and mitochondrial coupling (Fig 4F) [28,29]. Notably, $scaf1^{-/-}$ embryos were significantly more sensitive than $scaf1^{+/+}$ embryos to the inhibition of CI, CIII, and CIV, which participate in the formation of SCs, but were more resistant to the inhibition of CII (Fig 4F), which does not super-assemble. In addition, the inhibition of CV (ATP synthesis) and the coupling between the ETC and ATP synthesis with FCCP were both insensitive to the presence of Scaf1 (Fig 4F).

To gain direct insight on the impact of the Scaf1 ablation in mitochondrial respiration, we estimated the oxygen consumption capacity (OCR) of live mutant and wild-type zebrafish larvae using the Seahorse XFe24 analyzer [30]. Whereas the ablation of Scaf1 had no impact on the basal respiration, maximum oxygen consumption was significantly lower in zebrafish larvae lacking Scaf1 (Fig 4G and H). Notably, mitochondrial respiration of $scaf1^{-/-}$ larvae was more sensitive to antimycin and rotenone (Fig 4G), providing a plausible explanation for the higher lethality of these drugs on $scaf1^{-/-}$ larvae (Fig 4F).

The impact of Scaf1 ablation on mitochondrial respiration performance was also assessed in isolated mitochondria from adult zebrafish (Fig 4I–K). Both respiratory control ratio (RCR) and phosphate/oxygen ratio (P/O ratio) were unaffected by the loss of Scaf1 (Fig 4J and K). However, we again observed that maximum respiration was significantly lower in $scaf1^{-/-}$ zebrafish mitochondria than in wild-type counterparts (Fig 4I). We measured oxygen consumption in the presence of site I (pyruvate/glutamate and malate) or site II (succinate) substrates, finding that the decrease in maximum respiration was due to a decrease in site I substrate respiration, with site II respiration unaffected (Fig 4I). Additionally, we observed that the respiration of wild-type zebrafish mitochondria with the combination of site I and site II substrates was higher than that obtained with only site I or site II substrates. This phenomenon was described previously in mouse [1] and bovine mitochondria [31]. Of note, site II substrates alone allowed a respiration level similar to that with combined site I

and site II substrates in zebrafish $scaf1^{-/-}$ mitochondria, reproducing our observations in SCAF1-deficient mouse mitochondria [1].

These analyses demonstrate that the loss of Scaf1 has a direct functional impact on both OXPHOS performance and mitochondria structure.

### Diet determines $scaf1^{-/-}$ physiological phenotypes

The overall phenotype of $scaf1^{-/-}$ animals suggests metabolic inefficiency due to non-pathological decrease in mitochondria performance. We wondered whether increasing food availability could ameliorate the phenotype of $scaf1^{-/-}$ animals. We fed $scaf1^{+/+}$ and $scaf1^{-/-}$ fish with the double amount of food, which was distributed in more doses per day to ensure complete intake. Strikingly, this treatment was sufficient to eliminate the differences in growth (Figs 5A–C and EV5A–C) after only 4 weeks. As expected, $scaf1^{+/+}$ animals fed with double diet showed a prominent increase in adipose tissue area and adipocyte size, but these changes were not as evident in $scaf1^{-/-}$ animals that already presented elevated accumulation or fat at standard diet (Fig 5D–F). The increase in food also rescued female fertility in $scaf1^{-/-}$ fish (Fig 5G–I).

To understand the underlying adaptation of mutants and the positive effect of the double diet, we tested whether this phenomenon could be reproduced by feeding the animals with a diet with double the amount of fat (high-fat diet, HFD), which increases the caloric content to 141% (Appendix Table S1). Under HFD, the differences in growth and fertility induced by the lack of Scaf1 remained, showing that it is the double caloric intake mostly provided by proteins rather than fats which is responsible for rescuing growth and body weight in $scaf1^{-/-}$ animals (Fig 6A–H). HFD still led to an increase in adipose tissue area in zebrafish, being more prominent in $scaf1^{-/-}$ than $scaf1^{+/+}$, while there were no differences in adipocyte size (Fig 6I–L). Thus, the additional caloric intake upon HFD regime, instead of allowing restoration of normal growth, increments body fat in $scaf1^{-/-}$ zebrafish. This suggests that the rescue in size observed upon double diet regime is due to extra amount of protein content in the diet.

We next aimed to analyze whether diet could influence complex assembly or bioenergetics in the absence of Scaf1 (Fig 7). As expected, the comparative analysis of BNGE gels from mutant fish under double diet and standard revealed that double diet did not restore the super-assembly between CIII and CIV (Fig 7A).

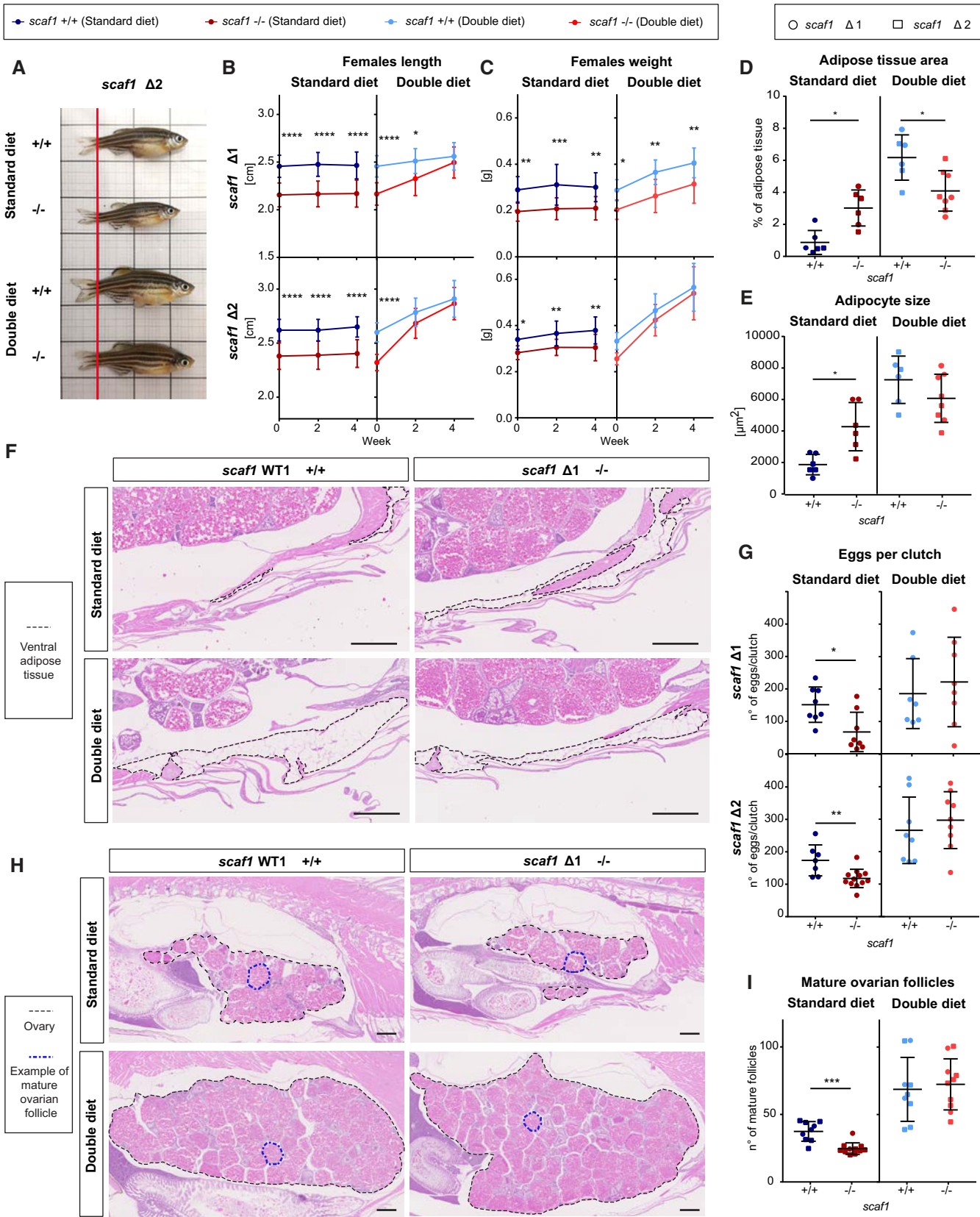

Figure 5.

◄

**Figure 5. Diet-induced recovery of $scaf1^{-/-}$ phenotypes.**

Data from females.

A    Representative images of $scaf1^{-/-}$ and $scaf1^{+/+}$ fish fed with the indicated diets.

B, C    Changes in (B) length and (C) weight over time ($\Delta1$ +/+ $n$ = 10, $\Delta1$ −/− $n$ = 10, $\Delta2$ +/+ $n$ = 10, $\Delta2$ −/− $n$ = 12–13).

D–F    Adipose tissue measurements on hematoxylin–eosin (H&E)-stained adult zebrafish sagittal sections. (D) Adipose tissue area per total section area (average of three sections/biological replicate) and (E) adipocyte size (average of 20–30 adipocytes of ventral adipose tissue per biological replicate; standard diet $\Delta1$ +/+ $n$ = 3, $\Delta1$ −/− $n$ = 3, $\Delta2$ +/+ $n$ = 3, $\Delta2$ −/− $n$ = 3, double diet $\Delta1$ +/+ $n$ = 3, $\Delta2$ +/+ $n$ = 3, $\Delta1$ −/−$n$ = 4, $\Delta2$ −/− $n$ = 4). $scaf1^{\Delta1}$ and $scaf1^{\Delta2}$ are represented with circles and squares, respectively. (F) Representative images of ventral fat deposits (dotted lines).

G    Number of eggs per clutch (standard diet $\Delta1$ +/+ $n$ = 8, $\Delta1$ −/− $n$ = 8, $\Delta2$ +/+ $n$ = 7, $\Delta2$ −/− $n$ = 12, double diet $\Delta1$ +/+ $n$ = 7, $\Delta1$ −/− $n$ = 8, $\Delta2$ +/+ $n$ = 8, $\Delta2$ −/− $n$ = 9).

H    Representative images of H&E-stained ovaries. Black dotted lines outline the ovaries, and blue dotted line indicates a mature follicle.

I    Quantification of mature ovary follicles per ovary section (average of three sections/biological sample; $\Delta1$ $n$ = 2–3, $\Delta2$ $n$ = 6).

Data information: (B, C) Two-way ANOVA, (D, I) unpaired $t$-test, and (E, G) one-way ANOVA. Data are represented as mean ± SD. *$P$ < 0.05, **$P$ < 0.01, ***$P$ < 0.001, ****$P$ < 0.0001. Scale bars = 500 μm.

Interestingly, the double diet significantly increased the maximum respiration (site I + II) capacity of isolated adult mitochondria from $scaf1^{-/-}$ animals, although it remained below wild-type respiration levels (Fig 7B). Therefore, despite the phenotype recovery, OXPHOS capacity in double diet is not fully normalized.

Since in the absence of $scaf1^{-/-}$ the ETC assembly is not recovered, the phenotypic rescue upon double diet must be underlying a compensatory metabolic adaptation. To gain molecular insight into this process, we performed RNAseq on skeletal muscle under the different diet regimes. Most of the OXPHOS genes were slightly downregulated in $scaf1^{-/-}$ muscle compared with $scaf1^{+/+}$ muscle, both in standard diet (Fig 7C) and in double diet (Fig 7D) conditions. In double diet, skeletal muscle showed an increase in the expression of OXPHOS genes both in $scaf1^{+/+}$ (Fig 7E) and $scaf1^{-/-}$ (Fig 7F) animals, although this increase was not sufficient to match OXPHOS gene expression levels of $scaf1^{+/+}$ animals (Fig 7D). Consistent with these findings, gene set enrichment analysis (GSEA) revealed that OXPHOS genes were downregulated in $scaf1^{-/-}$ when compared with wild-type muscle, independently of the diet, although it is always enhanced by double diet with respect to standard diet (Fig 7G). In contrast, ß-oxidation, TCA cycle, and catabolism of branched amino acids appeared downregulated only in standard diet (Fig 7G, Dataset EV2). Double diet induced a general upregulation of metabolic gene expression programs in $scaf1^{-/-}$ (Fig 7G). In agreement with the observed reduction in size and body weight, under standard diet $scaf1^{-/-}$ fish revealed changes in the enrichment scores of pathways related to growth and myogenesis, such as KRAS, IL6, JAK/STAT3 signaling [32] and E2F targets. In agreement with the diet-induced recovery of body length and size, upon double diet, IL6 and JAK/STAT3 and mitotic spindle pathways were higher in $scaf1^{-/-}$ compared to wild types and the myogenesis pathway was now presenting no differences between $scaf1^{-/-}$ and wild types (Fig 7H).

To sum up, the transcriptomic analysis suggests that double diet activates OXPHOS capacity accompanied by the enhancement of diverse metabolic pathways allowing the rescue of growth in $scaf1^{-/-}$ zebrafish in spite that the super-assembly of respiratory complexes and OXPHOS efficiency could not be restored.

### Food restriction in SCAF1-deficient mice mimics zebrafish phenotype

Given that C57BL/6 mice sub-strains, which harbor a non-functional SCAF1 (SCAF1[111/111]), lack any apparent phenotype, it was assumed that SCAF1 loss of function and CIII + CIV super-assembly have no bioenergetic relevance. The observation that $scaf1^{-/-}$ zebrafish do not show any evident phenotype when overfed and the fact that mice are regularly feed *ad libitum* prompted us to investigate whether the lack of functional SCAF1 in mice would reveal a higher sensitivity to long-term food restriction. Therefore, we analyzed weight gain during a period of starvation in C57BL/6 mice, SCAF1[−/−] mice harboring the full ablation of SCAF1, and SCAF1[113/113] mice carrying a functional SCAF1 gene [23]. Mice were fed *ad libitum* for 24 h every 3 days followed by a fasting period with restricted feeding regime for 12 weeks (Fig EV5D). During the first days, SCAF1[113] males were able to maintain or even increase their weight, suggesting an initial adaptive response (Fig EV5E). On the contrary, SCAF1[111] and SCAF1[−/−] males showed the opposite trend and lost rapidly weight (Fig EV5E). After this initial period, mice in all three groups started gaining weight. The sensitivity to weight changes under starvation was not observed in females (Fig EV5E), suggesting gender-specific mechanisms of adaptation to fasting in mice [33].

Overall, our results show that the effect of Scaf1 on organismal physiology is not restricted to zebrafish but is also shared in a mammalian model.

## Discussion

Here, we show that a lack of Scaf1-mediated CIII-CIV super-assembly impairs the bioenergetic efficiency of the ETC and leads to non-pathological physiological alterations at the organismal level.

Our findings show that overall the organization pattern of respiratory SC is well conserved among vertebrates. Nonetheless, zebrafish revealed some unique features. Compared to mammals, there is a higher proportion of super-assembly of the dimer CIV (IV$_2$), both in the form of SC III$_2$ + IV$_2$ and SC I + III$_2$ + IV$_2$ SCs. SCs containing dimers of CIII and CIV had also been described in yeast [34]. However, there, heterodimers are formed by SC III$_2$ + 2IV, instead of SC III$_2$ + IV$_2$, as observed here in the zebrafish. We also provide confirmation of the existence of the SC I + IV and SC I + IV$_2$ in vertebrates, for which there is only one report in mammals based on mass spectrophotometry [6]. The presence of this association is intriguing since the lack of CIII precludes a role in respiration. We suggest that these SC might

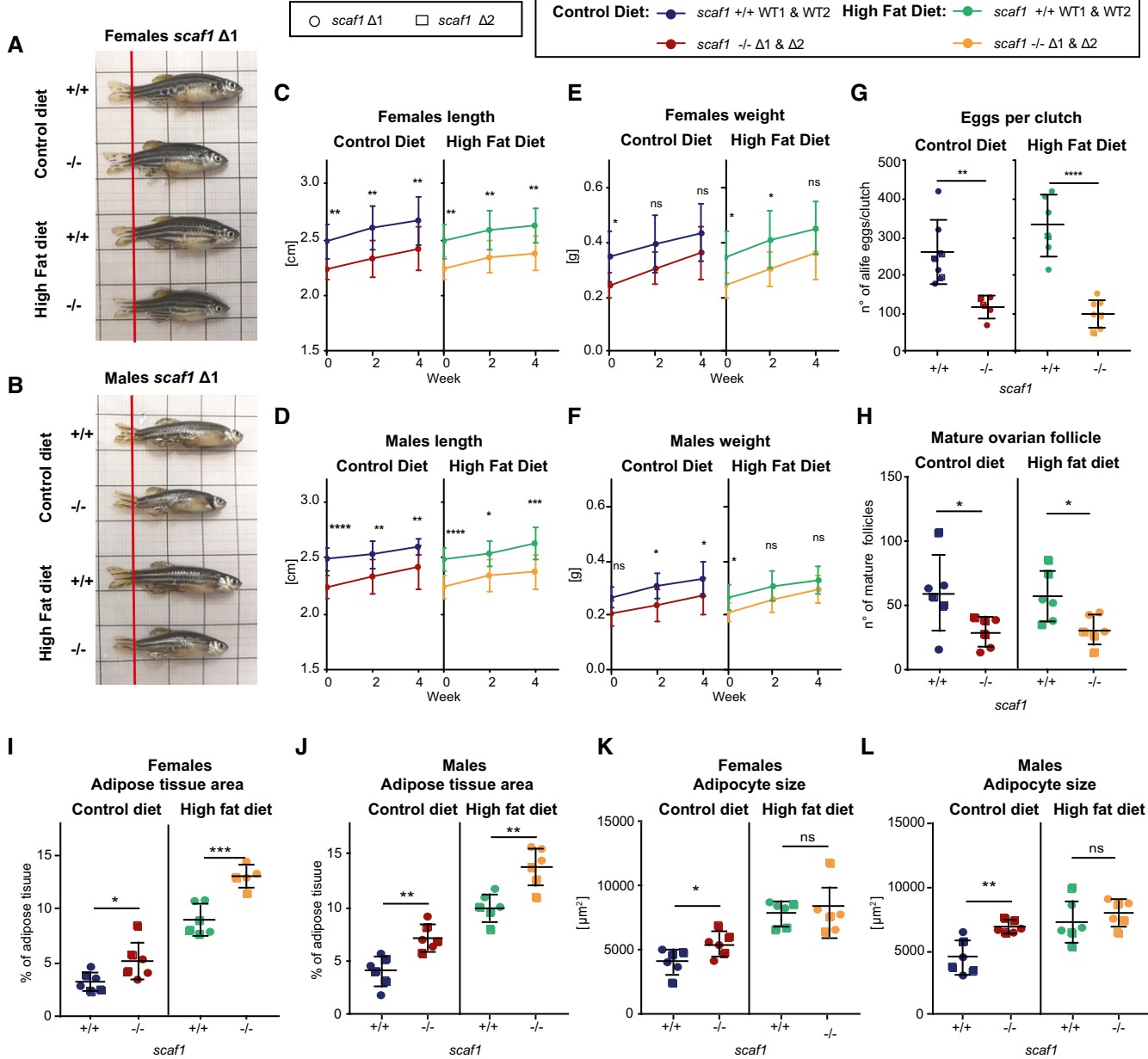

**Figure 6. Lack of recovery of Scaf1$^{-/-}$ phenotypes after high-fat diet.**

A, B  Representative images of scaf1$^{-/-}$ and scaf1$^{+/+}$ females (A) and males (B) fed with the indicated diets.

C, D  Length of females (C) and males (D) after the indicated diets ($\Delta 1$ +/+ n = 5, $\Delta 1$ −/− n = 5, $\Delta 2$ +/+ n = 5, $\Delta 2$ −/− n = 5).

E, F  Weight of females (E) and males (F) after the indicated diets ($\Delta 1$ +/+ n = 5, $\Delta 1$ −/− n = 5, $\Delta 2$ +/+ n = 5, $\Delta 2$ −/− n = 5).

G  Number of eggs per clutch (control diet: $\Delta 1$ +/+ n = 4, $\Delta 1$ −/− n = 3, $\Delta 2$ +/+ n = 3, $\Delta 2$ −/− n = 2, high fat diet: $\Delta 1$ +/+ n = 5, $\Delta 1$ −/−, n = 5 $\Delta 2$ +/+ n = 1, $\Delta 2$ −/− n = 2).

H  Quantification of mature ovary follicles per ovary section in hematoxylin–eosin (H&E) histological sections (average of three sections/biological sample; $\Delta 1$ n = 3, $\Delta 2$ n = 3).

I–L  Adipose tissue quantification in H&E sections ($\Delta 1$ n = 3, $\Delta 2$ n = 3) in females (I, K) and males (J, L): adipose tissue area (I, J) and adipocyte size (K, L).

Data information: (C–F) Two-way ANOVA. (G–L) Unpaired $t$-test. Data are represented as mean ± SD. ns $P > 0.05$, *$P < 0.05$, **$P < 0.01$, ***$P < 0.001$, ****$P < 0.0001$.

have a structural role. Interpretation of BNGE data should take into account the herein provided identification of SC I + IV SC without CIII. Indeed, given that SC I + IV$_2$ and SC I + III$_2$ have similar mass and co-migrate, they may be wrongly identified as

a respirasome. Finally, we describe here a truncated form of Scaf1, which we also found in the mouse, and that is generated by the proteolytic digestion by calpain 1 [23]. Again, its identification is crucial for correct interpretation of BNGE profiles. The

fact that this precise processing is observed both in zebrafish and mouse Scaf1 calls for further investigation of this phenomenon.

Zebrafish Scaf1 also acts as a SC assembly factor responsible for CIII-CIV interaction, and its role is therefore conserved in a non-mammalian vertebrate. Moreover, overnight fasting also leads to

**Figure 7.**

**Figure 7.   Molecular basis for diet-induced recovery of scaf1$^{-/-}$ phenotypes.**

A   Immunoblot of the indicated proteins of BNGE from female scaf1$^{+/+}$ and scaf1$^{-/-}$ whole zebrafish mitochondria for the indicated diet (representative of n = 2).

B   Maximum uncoupled (FCCP) oxygen consumption rate in whole zebrafish mitochondria (females Δ1 n = 4 and Δ2 n = 4, and same number for their respective controls) with glutamate (Glu), malate (Mal), and succinate (Succ).

C–G   RNAseq data from scaf1$^{+/+}$ and scaf1$^{-/-}$ skeletal muscle for the indicated diet (standard diet scaf1$^{+/+}$ and scaf1$^{-/-}$ n = 4, double diet scaf1$^{+/+}$ n = 3, scaf1$^{-/-}$ n = 4). (C-F) Volcano plots of differentially expressed genes (DEGs). (C) Comparison between scaf1$^{-/-}$ and scaf1$^{+/+}$ zebrafish in standard diet. (D) Comparison between scaf1$^{-/-}$ and scaf1$^{+/+}$ zebrafish in double diet. (E) Comparison of scaf1$^{+/+}$ zebrafish in double diet and standard diet. (F) Comparison of scaf1$^{-/-}$ zebrafish in double diet and standard diet. In blue, significant DEGs (Padj < 0.05, log$_2$FC > |1|); in gray, not significant DEGs; red circles represent non-significant differentially regulated OXPHOS genes, green circles represent significant differentially regulated OXPHOS genes, and purple circle represents scaf1 (cox7a2l). (G) Heatmap of metabolic pathways differentially regulated according to gene set enrichment analysis (GSEA) in the indicated comparisons.

H   Heatmap of differentially regulated growth hallmarks according to GSEA analysis in the indicated comparisons.

Data information: (G, H) White squares Padj > 0.05, colored squares Padj < 0.05. Color scales goes from blue (downregulated) to red (upregulated) gene sets. (B) T-test analysis. Data are represented as mean ± SD. *P < 0.05, **P < 0.01.

changes in OXPHOS super-assembly [1]. Thus, the *plasticity model* of mitochondrial ETC organization [35] is also valid in a fish model. While in the absence of Scaf1 the bands containing SC III$_2$ + IV$_{1-2}$ disappeared, a substantial amount of SC I + III$_2$ + IV remained. We suggest that, similar to what has been described in mammals [2,8, 23], in scaf1$^{-/-}$ animals SC I + III$_2$ + IV is preserved through the independent interaction of CI with CIII or CVI. However, the interaction between CIII and CIV within the SC I + III$_2$ + IV is lost due to the lack of Scaf1.

While the role of Scaf1 in supercomplex assembly is now accepted, its impact on mitochondrial bioenergetics is still not fully understood. A first study reported changes in substrate-dependent electron flux in mitochondria isolated from mouse liver from a SCAF1 mutant strain compared to a strain with a fully functional SCAF1 protein [1]. A second report on a human cellular model lacking SCAF1 could not confirm a change in mitochondrial bioenergetics [17]. However, a third study on human cells shows a change in the bioenergetic profile in *SCAF1* mutants upon stress [18]. Therefore, the role of SCAF1 on bioenergetics is, at this point, still disputed. Here, using the zebrafish as a whole animal model, we found that under physiological conditions, Scaf1 loss of function impacts bioenergetics in the substrate-dependent electron flux and maximum respiration capacity, corroborating a role of Scaf1 on mitochondrial bioenergetics.

The zebrafish model has allowed to assess the physiological role of Scaf1 and OXPHOS super-assembly at a whole organism level. Our assessment of the impact of Scaf1 ablation unequivocally demonstrates its physiological role and, by extension, a role of respiratory SCs in organismal physiology. However, we cannot exclude an unlikely possibility that the physiological consequences of Scaf1 loss of function are due to unknown functions independent on its role as a SC assembly factor. The lack of SC formation due to Scaf1 ablation reduces the efficiency of conversion of nutrients into energy, mimicking some features associated with malnutrition in zebrafish [24], such as impaired growth and fertility. It might seem paradoxical that there is an increase in adipose tissue deposits in normal diet-fed scaf1$^{-/-}$ zebrafish compared with controls. We hypothesize that the mild reduction of OXPHOS performance leads to an inefficient use of nutrients to promote tissue grow with the consequence of their storage in form of fat. In agreement, overfeeding protein content but not fat restored growth and fertility in scaf1$^{-/-}$ animals. Indeed, this rescue in organismal physiology occurs in the absence of a restored assembly between CIII and CIV and therefore requires metabolic adaptations to overcome the loss

of metabolic efficiency. Our transcriptomic data provide important first hints on the underlying mechanisms. The specific molecular and metabolic pathways and the determination of the key organs controlling metabolic adaptation merit further investigation. Importantly, our data also show that the effect of Scaf1 on weight gain is not restricted to zebrafish, but also becomes evident in mice models, when studied in a condition of caloric restriction.

In sum, we conclude that the incorporation of CIV into SCs through Scaf1 provides metabolic fitness allowing the organism to adapt to changing environmental energy supply or energy demands.

## Materials and Methods

### Zebrafish husbandry

Experiments were approved by the Community of Madrid "Dirección General de Medio Ambiente" in Spain and the "Amt für Landwirtschaft und Natur" from the Canton of Bern, Switzerland. All animal procedures conformed to EU Directive 86/609/EEC and Recommendation 2007/526/EC regarding the protection of animals used for experimental and other scientific purposes, enforced in Spanish law under Real Decreto 1201/2005. Experiments in Switzerland were conducted under the licenses BE95/15 and BE11/17. Experiments were conducted with adult zebrafish aged 5–10 months and raised at 13 fish per 2-l tank. Housing conditions were 28°C temperature, 650–700 µs/cm conductivity, and pH 7.5; 10% water exchange per day and lighting conditions were 14:10 h (light: dark). To guarantee a stable fish density between groups, occasionally scaf1$^{-/-}$, scaf1$^{+/-}$, or scaf1$^{+/+}$ fish were grown in the same tank as transparent Casper fish [36]. Standard feeding schedule was during weekdays: three times per day, once artemia (Ocean Nutrition) and twice dry food (ZM-000, Gemma Micron 150 and 300 for larvae, juveniles, and adults stages, respectively), during weekend: one time per day dry food. All zebrafish used had an AB genetic background, and original parental mice were purchased from ZIRC. The newly generated fish lines scaf1$^{Δ1}$ and scaf1$^{Δ2}$ are deposited in Zfin as cox7a3$^{brn1}$ and cox7a3$^{brn2}$, respectively. The transgenic line was deposited as Tg(ubi:scaf1)$^{brn3}$.

### Mouse husbandry

As with zebrafish, all animal procedures conformed to EU Directive 86/609/EEC and Recommendation 2007/526/EC regarding the

protection of animals used for experimental and other scientific purposes, enforced in Spanish law under *Real Decreto 1201/2005*. Four-month-old mice from the strains C57BL/6JOlaHsd and CD1 were used as a source for mitochondria purification of *scaf1*[111] and *scaf1*[113] mice, respectively. Original parental mice were purchased from Harlan Laboratories.

## Mouse generation

C57BL/6JOlaHsd mice with the functional version of SCAF1 were generated as previously described [2]. C57BL/6JOlaHsd mice knock out for SCAF1 were generated by microinjection of ES cells knock out in the first alleles from EuMMCR repository in a C57BL/6JOlaHsd blastocysts. Further, the blastocysts were implanted in a pseudopregnant female C57BL/6JOlaHsd.

## Protein sequences alignment

FASTA files of amino acids sequences of Scaf1 were obtained from the public NCBI databases for mouse *Cox7a2l*[113] and *Cox7a2l*[111]. The zebrafish paralog (named *cox7a2l* or *cox7a3*) was identified in Ensembl, and its amino acid sequence was aligned to mouse sequences using the Clustal Omega platform.

## Generation of Scaf1 loss of function fish and genotyping

CRISPR/Cas9 sgRNAs (sgRNA 1: GATCATAGCAGGGGATTCGG AGG, sgRNA2: GGAGTACATGGGTAAAAACA GGG) were designed using the CCTop tool [37], cloned in the plasmid MLM3636 (Addgene #43860), and synthesized and purified as described elsewhere [38].

The two sgRNAs were co-injected at 120–150 ng/µl together with homemade 6.5 µM Cas9 protein-produced from the pCS2-nCas9n plasmid (Addgene #47929) in NEB Cas9 buffer (NEB #B0386A) into zebrafish 1-cell-stage embryos.

Founder animals were identified at 3 months post-fertilization (mpf) by fin clip PCR analysis using the primers 5′TCCACTCT GCTTACTTCACAC3′ and 5′TTTGCTTTGTCTGTATGTCCTG3′ and were crossed with AB wild-type fish to generate F1 progeny. PCR products from the mutant allele of the F1 heterozygous were purified from gel bands (NEB #T1020S) and analyzed by Sanger sequencing. Two lines derived from different injection rounds and progenitors with two different deletions were established: *scaf1*[Δ1] and *scaf1*[Δ2] (deposited in Zfin as *cox7a3*[brn1] and *cox7a3*[brn2], respectively).

Genotyping during line maintenance used the described primers. All experiments were performed comparing *scaf1*[Δ1/Δ1] and *scaf1*[Δ2/Δ2] with their respective wild-type sibling lines coming from the same founder and AB mating. A maximum of four in-cross generations were used for the experiments.

## Generation of Scaf1 gain of function model in *scaf1*[−/−] fish

A transgenic line for *scaf1* gain of function was generated using the cDNA sequence of *scaf1* (*cox7a2l/cox7a3*) and the *ubiquitin* promoter (*ubi*) to drive ubiquitous expression: *Tg(ubi:scaf1)* (Fig EV4). The construct was generated by Gibson Assembly (NEB #E2611) of four fragments: iTol2Amp-Cryst:GFP cassette [39], *ubi* promoter from *ubi:switch* [40], *cox7a2l* cDNA and beta-globin intron

from *GFP-5xUAS-wt1bDN;cryaa:eCFP* [41]. Correct Gibson Assembly was corroborated by Sanger sequencing. The final purified plasmid was injected at 25 ng/µl into one-cell-stage embryos of *scaf1*[−/−] (*scaf1*[Δ1/Δ1]) together with 25 ng/µl Tol2 recombinase mRNA. F1 of heterozygous transgenic fish were compared with their siblings not positive for the transgenic construct and *scaf1*[+/+] fish born the same day.

## Mitochondria isolation

Whole fish or lateral skeletal muscle was cut into small pieces and rinsed using PBS and homogenization medium A (0.32 M sucrose, 1 mM EDTA, and 10 mM Tris–HCl, pH 7.4) at 4°C. Clean minced tissue was transferred to a manual Dounce tissue grinder containing homogenization medium A supplemented with fatty acid-free 0.1% bovine serum albumin (BSA-FFA; Sigma A7030). Occasionally, a motor-driven Teflon pestle with six up and down strokes at 650 rpm was used to replace the manual tissue grinder. The homogenate was centrifuged twice for 10 min at 800 $g$ and 4°C. The supernatant was then centrifuged for 10 min at 10,000 $g$ and 4°C. The pellet from this step was resuspended in 1 ml of MAITE buffer (0.25 mM sucrose, 75 mM sorbitol, 100 mM KCl, 0.05 mM EDTA, 5 mM $MgCl_2$, 10 mM Tris–HCl, and 10 mM orthophosphoric acid, pH.7.4) supplemented with 0.1% BSA-FFA and centrifuged for 10 min at 10,000 $g$ at 4°C. The clean pellet was resuspended in 0.5–1 ml MAITE + BSA-FFA buffer, and the concentration was quantified using the Bradford assay (Sigma-Aldrich). Mitochondria were stored at −80°C in MAITE + BSA-FFA buffer or centrifuged for 10 min at 10,000 $g$ and 4°C and subsequently resuspended in the buffer required for the following analysis. All steps were performed on ice.

For functional mitochondria assays, clean minced tissue was homogenized in homogenization buffer (67 mM sucrose, 50 mM Tris, 50 mM KCl, 10 mM EDTA, 0.1% BSA-FFA, pH 7.4) [42] and centrifuged twice for 10 min at 800 $g$ and 4°C. The supernatant was centrifuged 10 min at 8,000 $g$ and 4°C and washed with MAITE buffer as described above.

Mouse mitochondria were isolated from soleus skeletal muscle or liver as described [2], solubilized in 4 g/g digitonin and run in parallel with zebrafish samples.

## SDS and blue native gel electrophoresis

A total of 100–200 µg of whole fish mitochondria were resuspended in loading buffer (50 mM Tris–HCl pH 6.8, 2% SDS, 10% glycerol, 1% β-mercaptoethanol, 12.5 mM EDTA, 0.02% bromophenol blue) at a final concentration of 2 µg/µl, and 30 µg was loaded onto 12% hand-cast sodium dodecyl sulfate (SDS) acrylamide gels.

BNGE was performed as described [43] using 4 g/g digitonin-treated mitochondria; 30 µg of muscle or whole fish mitochondria or 20 µg of mouse soleus muscle or liver mitochondria was loaded onto 15 well 3–13% hand-cast native gels.

For 2D-BNGE, the first dimension used 125 µg whole-body zebrafish mitochondria treated with 4 g/g digitonin in 10-well 3–13% hand-cast native gels. The second dimension was performed in native 3–13% gels adding 0.02% DDM in the electrophoresis cathode buffer [43].

Two-dimensional denaturing electrophoresis (2D-BNGE/SDS–PAGE) was performed using 125 µg whole-body zebrafish

mitochondria treated with 4 g/g digitonin in 10-well 3–13% hand-cast native gels as the first dimension. The second denaturing dimension with SDS was performed as described [43].

## BNGE in-gel activity assays

After electrophoresis, BNGE gels were incubated for CI (0.1 mg/ml NADH and 2.5 mg/ml nitroblue-tetrazolium in 2 mM Tris–HCl pH 7.4 buffer) or CIV (1 mg/ml cytochrome $C$ and 0.5 mg/ml in 3,3′-diaminobenzidine in phosphate buffer pH 7.4) activity [44].

## Immunoblotting

Blue native gel electrophoresis or SDS gels were electroblotted onto Hybond-P-polyvinylidene fluoride (PVDF) membranes (Immobilon-FL, IPFL00010) and immunoblotted with antibodies against the different subunits of the OXPHOS complexes: anti-Ndufs3 mouse monoclonal (Abcam, AB14711), anti-Uqcrc2 rabbit polyclonal (Proteintech, 14742-1-AC), anti-Co1 mouse monoclonal (Thermo Fisher, 459600), and anti-Cox7a2l rabbit polyclonal (St John's laboratory, STJ110597). Anti-Vdac1 (Abcam, AB15895) was used as loading control.

Secondary antibodies used were anti-rabbit IgG (H+L) Alexa Fluor 680 conjugate (Life Technologies, A-21076) and anti-mouse DyLight 800 (Rockland, 610-145-121), and images were acquired with the ODYSSEY Infrared Imaging System (LI-COR). Immunoblotting of 1-dimension SDS using COX7A2L antibody was analyzed by the ECL detection system using polyclonal goat anti-rabbit (Dako, P0448) as the secondary antibody.

## Low-protein/low-fat diet experiment

One experiment was performed with eight wild-type fish at 7 mpf per group at a density of five fish/l. Transparent Casper fish [36] were used to reach equal tank density when necessary. Fish were randomly selected, and measured at the beginning of the experiment and assigned to one diet group. The control group was fed with Sparos control diet, and the experimental group was fed with Sparos LP/LF diet (Appendix Table S1) three times/day during weekdays and one time/day during weekends (15 mg/tank per dose). Fish were measured and sacrificed after 6 weeks. Mitochondria were freshly isolated in pools of two animals/sample.

## Blue-DiS proteomics

Three replicates of 200 μg of 4 g/g digitonin-treated whole fish mitochondria from $scaf1^{-/-}$ and $scaf1^{+/+}$ animals were run in a 3–13% hand-casted native gels, which were then stained with Coomassie Brilliant Blue R-250 and cut in 36 slices. The slices were processed and analyzed by data-independent mass spectrometry as previously described [2].

## Fish size, length, and fertility assessment

Anesthetized adult fish were measured in length with a millimetric ruler and weighed on a precision balance. Larvae and juvenile fish length was measured using ImageJ/Fiji from pictures obtained in a Nikon SMZ25 stereo microscope. For fertility assessment, one female was crossed with one male of the same genetic background

and eggs were collected 20 min after direct mating. The number of live eggs was manually counted. To analyze embryo development, animals were staged at one-cell stage. Embryo development was then checked at different time points. The number of embryos in a determined developmental stage was counted at each time point and represented as %.

## Zebrafish histology

Whole fish were fixed in 4% paraformaldehyde for 24 h, washed three times in PBS for 10 min, and decalcified in Immunocal (American MasterTech) at room temperature (RT) for 24 h. Tissues were dehydrated and embedded in paraffin blocks. Histological sections (7 μm) were used for hematoxylin–eosin staining. Three sagittal sections of representative areas per biological replicate were analyzed in ImageJ/Fiji, and the average was represented for each biological sample. Adipose tissue area was measured in the dorsal, ventral, visceral, and intramuscular areas, and summarized and divided by the total tissue area. The area of 20–30 adipocytes in the ventral fat was measured per biological sample.

## Transmission electron microscopy

Hearts from 5 mpf zebrafish were fixed with 2.5% glutaraldehyde (Agar Scientific, Stansted, Essex, UK) and 2% paraformaldehyde (Merck, Darmstadt, Germany) in 0.1 M Na-cacodylate-buffer (Merck), pH of 7.33. Samples were fixed for 24 h before further processing. They were then washed with 0.1 M Na-cacodylate-buffer three times for 5 min, post-fixed with 1% $OsO_4$ (Electron Microscopy Sciences, Hatfield, USA) in 0.1 M Na-cacodylate-buffer at 4°C for 2 h, and then washed in 0.05 M maleic acid (Merck, Darmstadt, Germany) three times for 5 min. Thereafter, samples were dehydrated in 70, 80, and 96% ethanol (Alcosuisse, Switzerland) for 15 min at RT. Subsequently, cells were immersed in 100% ethanol (Merck) three times for 10 min, in acetone (Merck) twice for 10 min, and finally in acetone-Epon (1:1) overnight at RT. Next, samples were embedded in Epon (Sigma-Aldrich, Buchs, Switzerland) and left to harden at 60°C for 5 days. Sections were produced with an ultramicrotome UC6 (Leica Microsystems, Vienna, Austria): Semithin sections (1 μm) were used for light microscopy and stained with a solution of 0.5% toluidine blue O (Merck); ultrathin sections (75 nm) were used for electron microscopy. The sections, mounted on 200-mesh copper grids, were stained with uranyl acetate (Electron Microscopy Sciences) and lead citrate (Leica Microsystems) with an ultrastainer (Leica Microsystems). Images were taken in a blinded manner using an FEI Tecnai Spirit electron microscope and analyzed on ImageJ/Fiji, also in a blinded manner.

## Mitochondrial DNA content

Genomic DNA was extracted from muscle and heart of 8 mpf fish with the DNeasy Blood & Tissue Kit (Qiagen). Three nanograms of DNA were used for qPCR using primers (nDNA: 5′ATGGGCTGGGCGATAAAATTGG3′, 5′ACATGTGCATGTCGCTCCCAAA3′; mtDNA: 5′CAAACACAAGCCTCGCCTGTTTAC3′, 5′CACTGACTTGATGGGGGAGACAGT3′) and method described before [45]. Data are represented with the formula 2*2(nDNA CT − mtDNA CT).

**Embryo OXPHOS toxicity**

Embryos (4 dpf) from in-crosses of $scaf1^{-/-}$ ($scaf1^{\Delta1/\Delta1}$ and $scaf1^{\Delta2/\Delta2}$) fish and their respective $scaf1^{+/+}$ counterparts were treated for 4 h at 28°C with 3–4 concentrations of OXPHOS inhibitors: CI, rotenone (Sigma-Aldrich, R8875); CII, 3-nitroproionic acid (Sigma-Aldrich, N5636); CIII, antimycin A (Sigma-Aldrich, A8674); $H^+$-ATPase oligomycin (Sigma-Aldrich, O4876); coupling inhibitor, carbonyl cyanide 4-trifluoromethoxyphenylhydrazone (FCCP; Sigma-Aldrich, C2920), in 0.05% ethanol E3 medium; and CIV, sodium azide (Sigma-Aldrich S2002) in E3 medium. Ten embryos were placed in a 12-well plate and treated in a final volume of 2 ml [29]. Nine biological replicates in three technical replicates per fish group, which make a total of 90 embryos per condition, were analyzed and represented as a proportion of survival. Cardiac arrest was used as lethality parameter.

**Whole embryo oxygen consumption rate**

Mitochondrial function was determined with an XFe24 extracellular flux analyzer (Seahorse Bioscience). OCR was measured in dechorionated 48 hpf embryos. Embryos were staged (same size was corroborated) and placed one per well on an islet capture microplate filled with E3 egg water. The plate was incubated in an incubator without $CO_2$ at 28°C for 30 min. After measuring baseline OCR as an indication of basal respiration, OCR was measured after an injection of 2 μM of FCCP to determine maximal respiration. Finally, 0.5 μM of antimycin A and 0.5 μM of rotenone were added to block mitochondrial respiration.

**Oxygen consumption in isolated mitochondria from adult fish**

Fresh isolated mitochondria (100 μg/ml) were analyzed using a Clark-type electrode (Oxygraph O2k; Oroboros Instruments, Innsbruck, Austria) at 28°C in MirO5 medium [46]. CI-dependent oxygen consumption (site I) was measured with 10 mM pyruvate or 10 mM glutamate and 5 mM malate. CII-dependent oxygen consumption (site II) was measured with 10 mM succinate and 0.2 μM rotenone. CI- and CII-dependent oxygen consumption (site I + II) was measured with the same concentrations of pyruvate or glutamate, malate, and succinate. The uncoupled state was reached with the inhibition of CV ($H^+$-ATPase) with 4 ng/ml oligomycin and a titration of FCCP in 0.1–0.5 μM intervals until reaching the stable maximum OCR. Respiration was stopped with 30 mM sodium cyanide. The RCR (State 3/State 4) and P/O ratio were measured at 28°C in a Clark-type electrode (Hansatech) with the aforementioned substrates plus 0.5 mM ADP. State 3 was calculated at the maximum OCR after ADP addition, and State 4 was calculated when ADP was consumed. The P/O ratio was calculated from the same measurement.

**Double diet experiments**

Three independent diet experiments were performed with 7–9 mpf fish control siblings, and $scaf1^{-/-}$ fish in each experiment were born the same day and were grown in the same conditions. Experiments one (starting at 8.5 mpf) and two (starting at 7 mpf) were performed with $scaf1^{\Delta2}$ and their respective $scaf1^{+/+}$ fish ($n = 10$ fish per experimental group, per experiment). Experiment three (starting at 7 mpf) was performed with $scaf1^{\Delta1}$ and their respective $scaf1^{+/+}$ fish ($n = 20$ fish

per experimental group). Fish were randomly selected, and length and weight were measured. They were then distributed in equal groups according to their measurements. Ten fish from mixed sex were placed per tank (five fish/l in 2 l-tanks) and assigned to a diet category. Standard diet (Appendix Table S1) (weekdays: two times/day dry food 15 mg/tank Gemma Micro 300, one time/day artemia, weekends: one time/day dry food 15 mg/tank Gemma Micro 300) and double diet (weekdays: four times/day dry food 15 mg/tank Gemma Micro 300, one time/day artemia, weekends: one time/day dry food 15 mg/tank and one time/day 30 mg Gemma Micro 300). Fish were maintained under these conditions for 6 weeks, measuring length and weight every 2 weeks during the four first weeks of the experiment. At week 5, fish were in-crossed, and the number of eggs laid per female was counted. Data from the three different experiments are represented together. Fish were sacrificed after 6 weeks. Fish from experiment two were used for histological analysis and BNGE. Fish from experiment three were used for histological analysis, BNGE, and RNAseq.

**High-fat diet experiments**

Two independent diet experiments were performed with 7 mpf fish, one with $scaf1^{\Delta1}$ and the other with $scaf1^{\Delta2}$ ($n = 10$ per experimental group, five females, five males). Fish were randomly selected, length and weight were measured, and they were distributed in equal groups according to their measurements. Ten fish from mixed sex were placed per tank (five fish/l in 2-l tanks) and assigned to a diet category. The control group was fed with Sparos control diet (14.1% fats). The high-fat diet group was fed with a customized diet by Sparos modified from their control diet containing double the amount of fats (30.1%; Appendix Table S1). Both groups were fed three times/day on weekdays and one time/day on weekends. Fish were maintained in these conditions for 6 weeks measuring length and weight every 2 weeks during the four first weeks of the experiment. At week 5, fish were in-crossed and the amount of eggs laid per female was counted. Data from $scaf1^{\Delta1}$ and $scaf1^{\Delta2}$ were plotted and analyzed together. Fish were sacrificed after 6 weeks.

**Food restriction in mice**

Starting from weaning, 4-week-old mice of the indicated genotype mice went through 48 h of fasting alternated with 24 h of controlled feeding, both with free access to water for 40 days. The weight was recorded at the end of each feeding phase.

**RNA isolation and sequencing**

Muscle from females $scaf1^{\Delta1}$ after 6 weeks on the diets (standard diet $scaf1^{+/+}$ and $scaf1^{-/-}$ $n = 4$, double diet $scaf1^{+/+}$ $n = 3$, $scaf1^{-/-}$ $n = 4$) was dissected and stored at −80°C. RNA was isolated using TRIzol and purified (Zymo RNA Clean & Concentrator kit). RNA purity was evaluated using the Agilent Fragment Analyzer and used for the bar-coded library generation. Libraries were sequenced on the Illumina HiSeq 2500 System.

**RNAseq bioinformatic analysis**

Sequencing reads were pseudo-aligned to *D. rerio* cDNA database (Ensembl build 11, release 94) using Kallisto [47] version 0.45.0.

Quality control was assessed using FastQC. Downstream analysis was performed in RStudio, and differential expression between design groups was tested using DESeq2 [48], with log2 fold change shrinkage. Volcano plots and heatmaps were generated using the base graphics and ggplot2 package.

*Danio rerio* Gene Stable IDs were translated to *Mus musculus* Gene Stable IDs using biomaRt package [49], and the overall gene expression was analyzed with GSEA [50] using the KEGG pathway, the Hallmarks collection, or a customized gene list as biological insight. Gene list from KEGG database was used for biogenesis of amino acids; Val, Leu, and Ile degradation; Arg and Pro metabolism; and glutathione metabolism (Dataset EV3). The list of genes used for the rest of pathways in Fig 7G is included in Dataset EV3. Growth pathways were analyzed using the Hallmark database (Dataset EV4). Only data with adjusted *P*-value and FDR < 0.05 were considered significant and represented.

### Quantification and statistical analysis

Randomization of samples and blinding was performed when needed, and it is indicated in the corresponding method detail. Image analysis was performed in a blind manner. Sample size is indicated in each figure legend. Normal distribution was tested using D'Agostino-Pearson omnibus and Shapiro–Wilk normality tests. Outliers were identified using the ROUT test ($Q = 1\%$), and when identified, they were not used in the statistical test. A *t*-test was used when comparing two groups, and one- or two-way analysis of variance (ANOVA) was used when more than two groups were analyzed. ANOVA multiple comparison was performed by Sidak's multiple comparison test when not indicated or Fisher's LSD test when indicated. Statistical parameters are specified in each figure legend. All data representations and statistical analyses were performed using GraphPad Prism 7.

## Data and code availability

RNAseq raw data and related information have been deposited in the Gene Expression Omnibus database with accession nos. GSE133487 (http://www.ncbi.nlm.nih.gov/geo/query/acc.cgi?acc = GSE133487). Proteomic data have been deposited at: http://www.peptideatlas.org/PASS/PASS01560. We deposited zebrafish line information at ZFIN: ZFIN ID: ZDB-TGCONSTRCT-200310-2, ZFIN ID: ZDB-ALT-200327-11 and ZFIN ID: ZDB-ALT-200327-12. Raw data leading to figures have been deposited at Mendeley with the following DOI https://doi.org/10.17632/rjhxf9wsc2.1; https://doi.org/10.17632/fzyyz6gm5n.1; https://doi.org/10.17632/f3nhz9rp28.1; https://doi.org/10.17632/t2fcsh6c27.1; https://doi.org/10.17632/f37md5b388.1; https://doi.org/10.17632/jfvcwx9677.1; https://doi.org/10.17632/7dkm6ss6c2.1.

**Expanded View** for this article is available online.

## Acknowledgements

We are grateful to M. M. Muñoz-Hernandez, Raquel Martínez de Mena, and Dr Concepción Jiménez for technical assistance; the Animal facility and Microscopy Units from CNIC and R. Baal, Anna Gliwa and María Galardi for fish husbandry; Sandra Nansoz from the Department of Intensive Care Medicine for use of the Oroboros equipment at the University of Bern and Inselspital; and Mojgan Masoodi for scientific discussion. Electron microscopy sample preparation and imaging were performed with help of Ines Marques, Beat Hänni and Benoît Zuber and devices supported by the Microscopy Imaging Center of the University of Bern. J.A.E. was supported by Spanish Ministry of Economy and Competitiveness, MINECO (SAF2015-65633-R), CIBERFES (CB16/10/00282), and HFSP (RGP0016/2018). N.M. was funded by the ERC starting grant 337703, HFSP (RGP0016/2018), and SNF 31003A-159721. FA was funded by Swiss National Science Foundation grant 320030_170062. J.V. was funded by MINECO (BIO2015-67580-P), by Carlos III Institute of Health-Fondo de Investigación Sanitaria (PRB3, IPT17/0019 - ISCIII-SGEFI/ERDF, ProteoRed), the Fundación La Marato TV3, and by "La Caixa" Banking Foundation (HR17-00247). The CNIC is supported by the Ministry of Economy, Industry and Competitiveness (MEIC) and the Pro-CNIC Foundation and is a Severo Ochoa Center of Excellence (MEIC award SEV-2015-0505).

## Author contributions

CG-P designed, performed, or assisted the experiments. SC designed some preliminary experiments, generated mouse samples and experiments, and performed some immune detections in Appendix Fig S2. EC, RM, and JV performed the Blue-DiS proteomic and, together with JAE, analyzed and interpreted the data. PH-A performed experiments in Fig 4J and K. SL and FA performed and supervised the experiments in Fig 4G and H.. MB performed the RNAseq bioinformatics analysis. XL designed and helped in the high-fat diet experiment and performed some preliminary histological sections, which lead to the final Fig 3. JAE, NM, and CG-P wrote the manuscript. All authors reviewed the manuscript. JAE, NM, and CG-P conceived the project, designed the experiments, and interpreted the results. NM, JAE and FA obtained funding for this work.

## Conflict of interest

The authors declare that they have no conflict of interest.

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
