## [Review Process File · EMBO Reports]

Scaf1 promotes respiratory supercomplexes and metabolic efficiency in zebrafish

Carolina García-Poyatos, Sara Cogliati, Enrique Calvo, Pablo Hernansanz-Agustín, Sylviane Lagarrigue, Ricardo Magni, Marius Botos, Xavier Langa, Francesca Amati, Jesús Vázquez, Jose Enriquez, and Nadia Mercader

DOI: 10.15252/embr.202050287

Corresponding author(s): Nadia Mercader (nadia.mercader@ana.unibe.ch) , Jose Enriquez (jaenriquez@cnic.es)

Review Timeline:

Submission Date:	24th Feb 20
Editorial Decision:	17th Mar 20
Revision Received:	13th Apr 20
Accepted:	28th Apr 20

Editor: Deniz Senyilmaz Tiebe

Transaction Report:

Dear Nadia,

Thank you for submitting your manuscript to EMBO Reports. Your manuscript has been reviewed at another journal. I have heard back from our arbitrating advisor, who looked at the revised manuscript and your response to the original referee concerns, whose comments are below. Having looked at everything carefully, I would like to invite a minor revision, before I can accept the manuscript.

- Please address the minor concerns of the arbitrating advisor.
- Please provide 3-5 keywords for your study. These will be visible in the html version of the paper and on PubMed and will help increase the discoverability of your work.
- Please fill out and include an author checklist as listed in our online guidelines (<https://www.embopress.org/page/journal/14693178/authorguide>)
- The reference format currently does not follow EMBO Reports guidelines. Please see <https://www.embopress.org/page/journal/14693178/authorguide#referencesformat> for further information.
- We noted that Fig 6L is called out, but the figure has 2 panel E's so only goes to K.
- There is a callout to Table 2, but there is no such table.
- We noted the following regarding Dataset EV Legends: The Tables have an incorrect nonenclature. Table S2 is small and could be included in an Appendix file, other it should be Table EV2. The others should be Dataset EV# files.
- We noticed that there are currently 7 supplemental figure files. Some could be (for technical reasons max 5) Expanded View figures, the rest should make up an Appendix with a table of contents.
- We realized that Figure S4 is in landscape format, which should be converted to portrait.
- Data and Code Availability section - RNAseq data (GSE133487) need to be made publicly available.
- Data and Code Availability section - please provide an URL for zebrafish line information at ZFIN.
- Fig 7D-F is missing labelling in the figure itself.
- Methods section needs to be renamed as Materials and Methods.
- Papers published in EMBO Reports include a 'Synopsis' to further enhance discoverability. Synopses are displayed on the html version of the paper and are freely accessible to all readers. The synopsis includes a short standfirst summarizing the study in 1 or 2 sentences that are provided by the authors and streamlined by the handling editor. I would therefore ask you to include your synopsis blurb - as well as 2-5 one-sentence bullet points that summarize the key findings of the paper.
- In addition, please provide an image for the synopsis. This image should provide a rapid overview of the question addressed in the study but still needs to be kept fairly modest since the image size cannot exceed 550x400 pixels.
- Our production/data editors have asked you to clarify several points in the figure legends (see attached document). Please incorporate these changes in the attached word document and return it with track changes activated.

Referee #1:

This report investigates the role of SCAF1 in the assembly of respiratory supercomplexes in Zebrafish. Scaf1 is known to coassemble with respiratory complexes and that scf1 mutant mice display altered distribution of complexes into supercomplexes. However, these mice do not show overt physiological phenotype, raising the question of the functional importance of these supercomplexes.

Here, the authors generate scf1-null zebrafishes and observe that, in addition to a supercomplex assembly defect, they also have a metabolic phenotype, being smaller and accumulating fat, altogether indicating a respiratory deficiency. Interestingly, most phenotypes can be suppressed by doubling food doses. This prompted them to reevaluate the scf1 mutant and knock-out mice in condition where food is limiting, and here again they evidence a physiological phenotype. Altogether they find that scf1 plays an important physiological role, most likely through promoting the assembly of supercomplexes.

The authors responded adequately to the reviewers concerns. Many of the points of reviewer 1 were not addressed, but their bearing on the story is unclear (for many experiments proposed, whatever outcome would not affect the main conclusions of the paper). Reviewer 2 raises a tricky but important point. While the data hereing definitely indicate a physiological role for scf1, they do not ensure that this effect is related to the supercomplex assembly defect observed in the same mutants. The way to definitely test this causality inference would be to somehow rescue supercomplex assembly in the absence of Scaf1. Since there is no easy way to do it and since the causality, while unproven, is still very likely for reasons the authors explain clearly in the manuscript and their rebuttal, this should ne be a prerequisite for publication. A sentence in the discussion stating that while it is most likely that scf1 affects physiology via the assembly of supercomplexes, one cannot, at this stage, exclude the unlikely possibility that it does so through other mechanisms.

Minor points

p8 lines 165-169. Could be rephrased. The appearance of greater number of band is not due to the discontinuation of the antibody, but to the use of a substitute. And this has nothing to do with mammal vs zebrafish, as the first sentence implies.

Figure 7, there is a wildtype control for both delta1 and delta2 alleles. these are labelled delta1 scf1+ and the mutants delta1 scf1- which is very misleading. Why not WT1 and WT2 versus delta1 and delta2?

Referee #1:

This report investigates the role of SCAF1 in the assembly of respiratory supercomplexes in Zebrafish. Scaf1 is known to coassemble with respiratory complexes and that scaf1 mutant mice display altered distribution of complexes into supercomplexes. However, these mice do not show overt physiological phenotype, raising the question of the functional importance of these supercomplexes.

Here, the authors generate scaf1-null zebrafishes and observe that, in addition to a supercomplex assembly defect, they also have a metabolic phenotype, being smaller and accumulating fat, altogether indicating a respiratory deficiency. Interestingly, most phenotypes can be suppressed by doubling food doses. This prompted them to reevaluate the scaf1 mutant and knock-out mice in condition where food is limiting, and here again they evidence a physiological phenotype. Altogether they find that scaf1 plays an important physiological role, most likely through promoting the assembly of supercomplexes.

The authors responded adequately to the reviewers concerns. Many of the points of reviewer 1 were not addressed, but their bearing on the story is unclear (for many experiments proposed, whatever outcome would not affect the main conclusions of the paper). Reviewer 2 raises a tricky but important point. While the data hereing definitely indicate a physiological role for scaf1, they do not ensure that this effect is related to the supercomplex assembly defect observed in the same mutants. The way to definitely test this causality inference would be to somehow rescue supercomplex assembly in the absence of Scaf1. Since there is no easy way to do it and since the causality, while unproven, is still very likely for reasons the authors explain clearly in the manuscript and their rebuttal, this should ne be a prerequisite for publication. A sentence in the discussion stating that while it is most likely that scaf1 affects physiology via the assembly of supercomplexes, one cannot, at this stage, exclude the unlikely possibility that it does so through other mechanisms.

We thank the reviewer very much for this evaluation. We have included a sentence in the discussion pointing out the unlikely possibility of SC assembly independent functions of SCAF1:

“However, we cannot exclude an unlikely possibility that the physiological consequences of Scaf1 loss of function are due to unknown functions independent on its role as a SC assembly factor.”

Minor points
p8 lines 165-169. Could be rephrased. The appearance of greater number of band is not due to the discontinuation of the antibody, but to the use of a substitute. And this has nothing to do with mammal vs zebrafish, as the first sentence implies.

We changed the sentence:” We found that this is due to the use of new Scaf1 antibody generated against the entire protein, as the previously well-established antibody generated against a Scaf1-specific peptide was discontinued.

Figure 7, there is a wildtype control for both delta1 and delta2 alleles. these are labelled delta1 scaf1+ and the mutants delta1 scaf- which is very misleading. Why not WT1 and WT2 versus delta1 and delta2?

We changed the labelling according to the request.

Dear Nadia,

Thank you for submitting your revised manuscript. I have now looked at everything and all looks fine. Therefore I am very pleased to accept your manuscript for publication in EMBO Reports.

Congratulations on a nice study!

Before we can transfer your manuscript to our production team, we need to sort out a couple of minor points.

1. Our data editors have asked you to clarify a couple of points in the legend of the newly added data (please see attached). Please address those with 'track changes on' in the attached word file. You can return the file to me per email.

2. I have taken the liberty of performing some minor changes in the items below to increase clarity/accessibility. Please take a look and confirm, or feel free to suggest further changes.

Synopsis: This study reveals a physiological function of mitochondrial respiratory supercomplex assembly in metabolism by analyzing *scf1* mutant zebrafish that are unable to form supercomplexes.

Bullet points:

- Comprehensive characterization of zebrafish respiratory supercomplexes reveals presence of supercomplexes similar to those of mice, as well as zebrafish specific ones.
- Null mutants for the super-assembly factor *scf1* lack CIII+ CIV supercomplex
- *scf1* null mutant zebrafish exhibit impaired mitochondrial bioenergetics
- *scf1* null mutants are smaller, weigh less and display reduced fertility
- Phenotypic consequences of *scf1* mutation can be rescued through by doubling food supply.

Abstract:

The oxidative phosphorylation (OXPHOS) system is a dynamic system in which the respiratory complexes coexist with super-assembled quaternary structures called supercomplexes (SCs). The physiological role of SCs is still disputed. Here we use zebrafish to study the relevance of respiratory SCs. We combined immunodetection analysis and deep data-independent proteomics to characterize these structures and found similar SCs to those described in mice, as well as previously unidentified SCs including III₂+IV₂, I+IV and I+III₂+IV₂. To study the physiological role of SCs, we generated two null allele zebrafish lines for supercomplex assembly factor 1 (*scf1*). *scf1*^{-/-} fish display altered OXPHOS activity due to the disrupted interaction of complex III and IV. *scf1*^{-/-} fish are smaller in size and show abnormal fat deposition and decreased female fertility. Doubling the food supply rescues these physiological phenotypes, which in parallel improves bioenergetics and alters the metabolic gene expression program without affecting SC assembly. These results reveal that SC assembly by *Scf1* modulates OXPHOS efficiency and ensures the optimization of metabolic resources.

Kind regards,

Deniz

Corresponding Author Name: Nadia Mercader & Jose Antonio Enriquez

Journal Submitted to: EMBO REPORTS

Manuscript Number: EMBOR-2020-50287V1